# Time-resolved mitochondrial screen identifies regulatory components of oxidative metabolism

Marcos Zamora-Dorta[1,9], Sara Laine-Menéndez[1,9], David Abia [1], Pilar González-García [2,3,4], Luis C López [2,3,4,5], Paula Fernández-Montes [6], Enrique Calvo[6], Jesús Vázquez [6], José Antonio Enríquez [6,7] & Eduardo Balsa [1,8✉]

## Abstract

**Defects in mitochondrial oxidative metabolism underlie many genetic disorders with limited treatment options. The incomplete annotation of mitochondrial proteins highlights the need for a comprehensive gene inventory, particularly for Oxidative Phosphorylation (OXPHOS). To address this, we developed a CRISPR/Cas9 loss-of-function library targeting nuclear-encoded mitochondrial genes and conducted galactose-based screenings to identify novel regulators of mitochondrial function. Our study generates a gene catalog essential for mitochondrial metabolism and maps a dynamic network of mitochondrial pathways, focusing on OXPHOS complexes. Computational analysis identifies RTN4IP1 and ECHS1 as key OXPHOS genes linked to mitochondrial diseases in humans. RTN4IP1 is found to be crucial for mitochondrial respiration, with complexome profiling revealing its role as an assembly factor required for the complete assembly of complex I. Furthermore, we discovered that ECHS1 controls oxidative metabolism independently of its canonical function in fatty acid oxidation. Its deletion impairs branched-chain amino acids (BCAA) catabolism, disrupting lipoic acid-dependent enzymes such as pyruvate dehydrogenase (PDH). This deleterious phenotype can be rescued by restricting valine intake or catabolism in ECHS1-deficient cells.**

**Keywords** Mitochondria; OXPHOS; CRISPR Screening; RTN4IP1; ECHS1
**Subject Categories** Metabolism; Organelles

## Introduction

Mitochondria are unique and complex organelles that perform essential functions in many aspects of cell biology (Monzel et al, 2023). Arguably, the most extensively understood role of the mitochondria, and the reason it is often referred to as the "powerhouse of the cell", is its function in cellular energy production through the oxidative phosphorylation system (OXPHOS) (Vafai and Mootha, 2012). This bioenergetic process necessitates the transfer of electrons to molecular oxygen through the mitochondrial electron transport chain (ETC), which involves four multi-subunit complexes (known as complex I–complex IV) and two mobile electron carriers, ubiquinone (also known as coenzyme Q10) and cytochrome c (cyt c). As a result of the respiratory chain's activity, a transmembrane proton gradient is generated, which complex V (or ATP synthase) harnesses to produce ATP (Enríquez, 2016). The OXPHOS system, central to cellular energy production, is intricately intertwined with the tricarboxylic acid (TCA) cycle, where diverse metabolic pathways, including glucose, glutamine, fatty acids, and amino acids, converge to supply substrates. This nexus intimately connects nutrient intake to mitochondrial energy generation. Mutations that compromise these critical metabolic pathways are the root cause of a heterogeneous group of genetically inherited disorders referred to as mitochondrial diseases, for which no cures are currently available (DiMauro and Schon, 2003; Thorburn, 2004).

A greater understanding of mitochondrial genetics has led to improved diagnosis over the past few years. Nonetheless, a significant number of patients with biochemically established mitochondrial diseases lack mutations in known mitochondrial disease genes, implying the existence of unidentified proteins involved in OXPHOS function (Legati et al, 2016). In addition, there have been numerous reported cases of mutations in genes whose function is not directly linked to OXPHOS, suggesting that these genes might have additional, unknown functions related to oxidative metabolism (Rodenburg, 2011). Even though we now possess a very detailed picture of the core proteins that comprise the OXPHOS complexes, the list of assembly factors and regulatory components is far from being completed. More surprisingly, it has been estimated that nearly 15% of the mitochondrial proteome lacks precise functional annotation (Rensvold et al, 2022; Pagliarini et al, 2008). Thus, a complete inventory of genes that are required for the proper function of the OXPHOS system would provide a molecular framework for the investigation of mitochondrial biology and pathogenesis.

[1]Departamento de Biología Molecular and Centro de Biología Molecular Severo Ochoa (UAM-CSIC), Madrid, Spain. [2]Instituto de Investigación Biosanitaria Ibs, 18016 Granada, Spain. [3]Departamento de Fisiología, Facultad de Medicina, Universidad de Granada, 18016 Granada, Spain. [4]Instituto de Biotecnología, Centro de Investigación Biomédica, Universidad de Granada, 18016 Granada, Spain. [5]Centro de Investigación Biomédica en Red Fragilidad y Envejecimiento Saludable (CIBERFES), 18016 Granada, Spain. [6]Laboratory of Functional Genetics of the Oxidative Phosphorylation System, Centro Nacional de Investigaciones Cardiovasculares Carlos III (CNIC), 28029 Madrid, Spain. [7]CIBER de Fragilidad y Envejecimiento Saludable, Instituto de Salud Carlos III, 28029 Madrid, Spain. [8]Instituto Universitario de Biología Molecular - IUBM (Universidad Autónoma de Madrid), Madrid, Spain. [9]These authors contributed equally: Marcos Zamora-Dorta, Sara Laine-Menéndez. ✉E-mail: eduardo.balsa@uam.es

It is widely reported that OXPHOS-deficient cells are viable in glucose-rich medium but cannot survive when glucose is replaced with galactose (Balsa et al, 2020). While this selective lethality has been exploited in the past to perform genetic screens, these attempts were not particularly tailored to identify regulatory components, but rather genes with strong phenotypes (Bayona-Bafaluy et al, 2011; Arroyo et al, 2016). This limited success may be attributed to several factors. First, the use of RNAi technology often results in only partial knockdown, necessitating complete knockdown to produce a phenotypic consequence (Le Vasseur et al, 2021). Second, the utilization of large genome-wide libraries makes it particularly challenging to detect the depletion of a gene in a highly complex library when the fitness effect is modest between the two conditions being compared (Thomas et al, 2021). Third, all of these studies focused solely on one specific time point, and it is likely that understanding temporal dynamics regarding the phenotype is necessary to capture a wide spectrum of gene candidates. Consequently, the identification of OXPHOS essential genes with pronounced phenotypes may overshadow the subtle effects of potential assembly factors and regulatory components that exhibit more modest phenotypes.

To overcome these challenges, we have generated a small CRISPR loss-of-function library exclusively containing all the nuclear-encoded mitochondrial genes (Mito-library). Subsequently, we carried out a galactose-based screening at multiple time points to uncover new regulators controlling OXPHOS function and mitochondrial metabolism. We assembled a list of genes vital for mitochondrial oxidative metabolism and developed a dynamic timeline that delineates the extensive network of components within mitochondrial pathways, particularly focusing on the OXPHOS complexes. Employing MAGeCK and clustering (*clust*) computational analyses, we identified two poorly characterized genes, RTN4IP1 and ECHS1, which exhibited significant correlation with established components of OXPHOS. RTN4IP1 was found to be critical for mitochondrial respiration and the assembly of complex I, while ECHS1 was revealed to regulate oxidative metabolism independently of its typical role in fatty acid oxidation. Deficiency in ECHS1 results in the buildup of toxic intermediates of BCAA metabolism, which adversely affect enzymes dependent on lipoic acid. Overall, we have demonstrated that conducting temporal dynamic screenings using our Mito-library is a robust approach to identify novel uncharacterized genes controlling the function and integrity of OXPHOS and mitochondrial metabolism.

# Results

## Galactose-based CRISPR screening using a custom-developed Mito-library

We postulate that genome-wide galactose-based screenings conducted exclusively at very early time points may lack the capacity to discern regulatory components of mitochondrial oxidative metabolism, which may manifest as more subtle, long-term phenotypes (Fig. 1A). To overcome these intrinsic constraints, we developed a new lentiviral library containing all the nuclear-encoded mitochondrial genes cataloged in human MitoCarta 3.0 (Rath et al, 2021). A total of 1124 genes were targeted with a coverage of four sgRNAs per gene. HeLa and 143B cell lines constitutively expressing the Cas9 nuclease were infected with the Mito-library, cultured in glucose-rich medium, and selected with appropriate antibiotics for at least 7 days to allow for CRISPR-mediated gene knockout. Next, cells were seeded in medium containing either glucose or galactose as the sole sugar source. Finally, cells were harvested immediately after selection (t0h) and at multiple time points (t24h, t48h, t1week, and t2weeks), and genomic DNA (gDNA) was isolated for sequencing to calculate the difference in abundance of each sgRNA between the initial seeding and final populations (Fig. 1B). We calculated log2 fold change values for the abundance of sgRNAs between galactose and glucose conditions (gal/glu ratio) at each time point for every gene relative to the initial population (t0h). A strong correlation between cell lines at the earliest time points was observed (24 h, $r = 0.7711$ and 48 h, $r = 0.7663$), which decreased over time (1 week, $r = 0.5612$ and 2 weeks, $r = 0.2973$) (Fig. EV1A). This could indicate cell-specific behaviors in multiple genes. Significant changes in genes ($p < 0.05$) for each time point were categorized as mild or severe according to their phenotype severity (mild: genes exhibiting a phenotype with a drop in fold change in the gal/glu ratio above 20% but below 35%. Severe: genes exhibiting a phenotype greater than 35%). For both mild and severe conditions, we observed a trend where genes with significant phenotypes increased over time, reaching a maximum at 1 week. This progressive increase was more accentuated in the category of genes with a severe phenotype, and overall suggests that incorporating a temporal resolution feature in this type of screening enhances the total number of genes detected (Fig. 1C). Notably, differences were observed in the total number of positive genes detected between HeLa and 143B cells, with the screening in HeLa cells yielding more positive genes and exhibiting a greater magnitude of the phenotype. Intriguingly, the percentage of genes exhibiting phenotypes begins to drop at 2 weeks. This observation suggests an accumulation of residual cells with low Cas9 activity, resulting in phenotypic behavior resembling wild-type cells due to the absence of gene editing. As expected, GO term analysis performed with genes exhibiting a phenotype identified pathways associated with cellular respiration, OXPHOS and ETC. However, we observed time-dependent variations when comparing early and late time points. This analysis revealed that mitochondrial translation and gene expression were the more significative pathways at early time points (24 and 48 h), while mitochondrial respiratory chain complex assembly emerged as the top pathway during later time points (1 and 2 weeks) (Fig. 1D). Collectively, this data indicates that employing small focused libraries in combination with multiple time points is a powerful strategy to uncover the temporal evolutions of phenotypic hierarchies, potentially enabling the identification of dynamic changes with subtle differences.

## Temporal profiling of mitochondrial pathways essential for oxidative metabolism

We proceeded to analyze the data using the terms defined by MitoCarta 3.0, which established well-defined ontologies and sub-ontologies, most of which are incontrovertible. Nevertheless, many are unavoidably redundant, as bifunctional proteins are necessarily assigned to multiple ontologies and some missadscriptions to particular ontologies are documented. All the genes in our Mito-

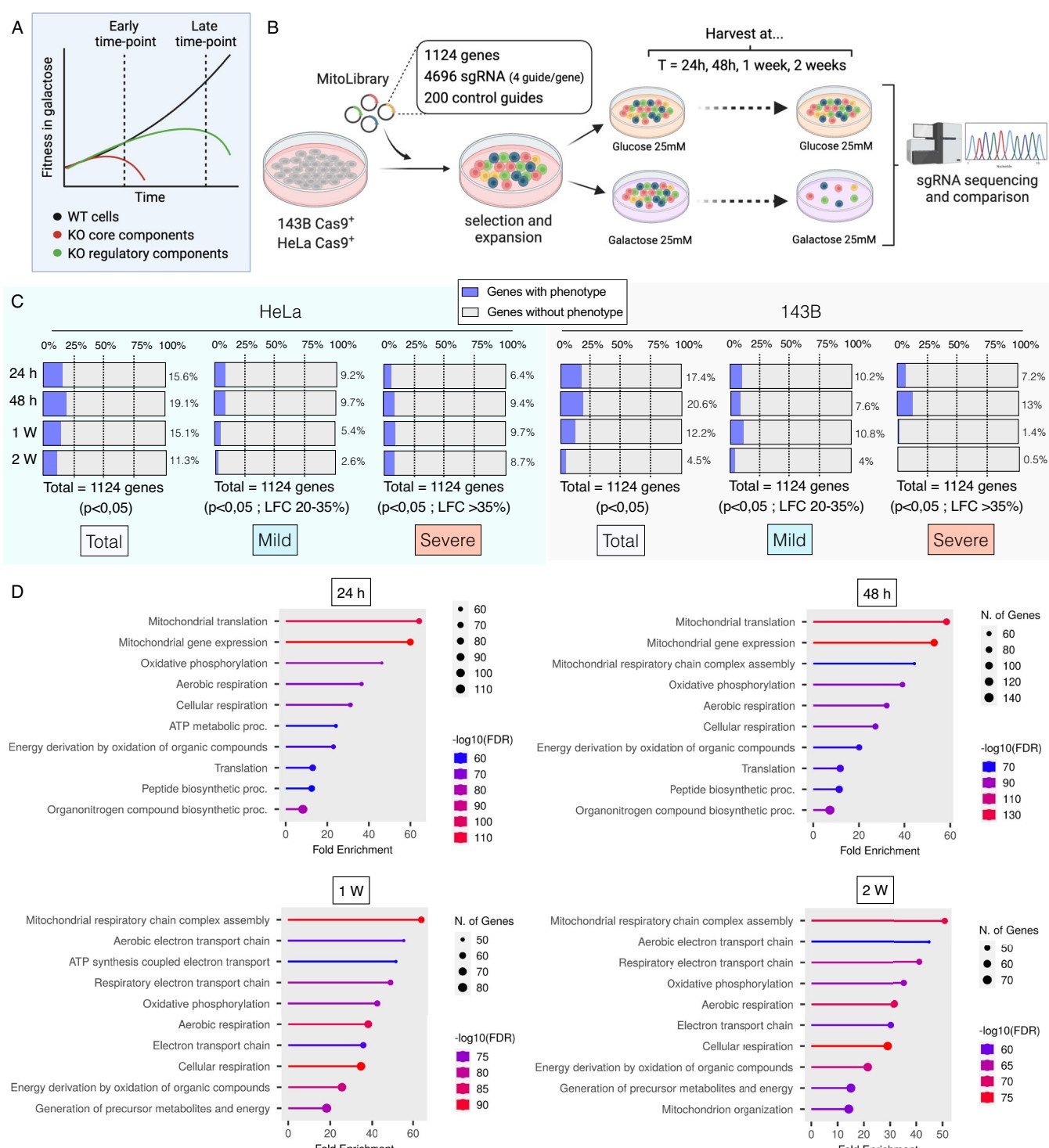

library were subdivided into the following functional categories: Dynamics and surveillance, Signaling, Mitochondrial central dogma (encompassing mtRNA translation, mtDNA maintenance, and mtRNA metabolism), Transport (encompassing protein import and small molecule transport), OXPHOS and metabolism; and Sankey diagrams were chosen as a visualization tool to depict the temporal dynamics of the phenotypic variations (Dataset EV1).

Dynamics and surveillance, along with the Signaling pathway, exhibited notable enrichment in genes with no discernible phenotype, implying that genes within these categories may be dispensable for proliferation in galactose, likely owing to their limited influence on OXPHOS activity (Fig. 2A,B). The only genes displaying a severe phenotype were ATP5ME, CYCS, and SLC25A5, which also possess redundant functions allocated in

**Figure 1. Galactose-based CRISPR screening using a custom-developed Mito-library.**

(A) Scheme illustrating the hypothesis that while OXPHOS core components will yield a strong death phenotype under galactose medium at early time points, regulatory OXPHOS components will yield a more modest death phenotype that becomes detectable at later time points. (B) Schematic overview of the mitochondrial-focused CRISPR-Cas9 screening approach. HeLa and 143B cells were transduced with a Mito-library of sgRNAs targeting ~1124 genes with a coverage of four sgRNAs per gene, and 200 additional control sgRNAs. After selection, transduced cells were cultured either in glucose or galactose for 24 h, 48 h, 1 week and 2 weeks. Cells were collected and their genomic DNA (gDNA) was isolated. sgRNA abundance was determined by deep sequencing, and final sgRNA counts were compared to initial counts to calculate the median differential score for each gene and the fold change between galactose and glucose conditions (gal/glu ratio) for each gene and time point. (C) Temporal variation of the total genes that scored positive in the screening across both cell types, and the distribution of these genes categorized by either mild or strong phenotypes. It is considered a positive phenotype when a drop in fold change in the gal/glu ratio is above 20%, being mild phenotype between 20% and 35%, and a severe phenotype above 35%. A p value <0.05 was considered statistically significant for all cases. The analysis, including statistical significance, was performed using the Broad Institute MaGECK methods. This hypergeometric tool calculates p values by assessing the consistency of ranks among guides targeting the same gene. (D) Gene Ontology (GO) enrichment analysis on the positive genes identified in the screening, showcasing the most enriched GO terms and their temporal variation over time. Source data are available online for this figure.

other categories (Fig. EV2A,B). Several genes associated with the mitochondrial central dogma exhibited both mild and severe phenotypes beyond the 24-h time point, particularly those involved in translation (Fig. 2C). We also detected several genes involved in mtDNA maintenance and mtRNA metabolism, mutations in which are well-known to cause mitochondrial diseases in humans. Examples of these include TWNK, LRPPRC, POLRMT, and MTO1 (Mootha et al, 2003; Ghezzi et al, 2012) (Fig. EV2C). The transport pathway also exhibited mild and severe phenotypes in a time-dependent manner (Fig. 2D), exemplified by the newly discovered mitochondrial NAD$^+$ transporter SLC25A51 and the ADP/ATP carrier SLC25A5, both of which play pivotal roles in oxidative metabolism (Fig. EV2D). Notably, we identified SFXN4 as one of the genes exhibiting a severe phenotype, consistent with recent studies demonstrating that, unlike SFXN1/2/3, which are involved in serine transport (Kory et al, 2018), SFXN4 is necessary for the formation of the ND2 module of complex I (Jackson et al, 2022) (Fig. EV2D). A significant number of genes within the Metabolism pathway exhibited a severe phenotype (Fig. 2E), particularly those involved in carbohydrate and lipid metabolism. However, the majority of these genes are primarily associated with the biogenesis of iron-sulfur clusters, cofactors such as lipoic acid, and metallochaperones (Fig. EV2E). Undoubtedly, OXPHOS was the category with the most enriched genes displaying a positive phenotype (Fig. 2F). As expected, core components of the ETC complexes manifested phenotypic changes before the assembly factors. Exceptions to this pattern were observed for factors involved in the early assembly of CI (NDUFAF5, NDUFAF4, or NDUFAF8) and CIV (COX17, COA6, or COX16), which were detected as early as 24 h. Late detection of assembly factors with severe phenotype includes genes such as NDUFAF3, SURF1, TACO1, SCO1, LYRM7, or TIMMDC1, whose mutations are commonly found in patients with mitochondrial diseases (Fernandez-Vizarra and Zeviani, 2021) (Fig. EV2F).

## High-resolution dynamic timeline illustrating the evolving behavior of OXPHOS components

Galactose medium is commonly utilized to uncover defects in the OXPHOS system. In fact, at each time point, OXPHOS genes account for a significant percentage of the total genes with phenotype, indicating that this screening setup was particularly successful in detecting genes that impact the OXPHOS system (Fig. EV3A). Given our effectiveness in capturing OXPHOS-related

genes, we proceeded with additional analysis focusing on the temporal fitness of each OXPHOS complex. We systematically divided each OXPHOS complex into its core subunits and assembly factors, tracking their phenotypic changes over time. In line with our initial hypothesis (Fig. 3A), we observed that the phenotype of core subunits tended to be more pronounced and occurred at earlier time points compared to assembly factors. Curiously, the only complex that did not follow this trend was complex CIV. We reasoned that this could be attributed to the presence of tissue-specific subunit paralogs for CIV (Cogliati et al, 2016). Consequently, we repeated the analysis, categorizing CIV genes into two groups: those with paralogs and those without. The CIV paralog subunits did not show a phenotype, whereas those lacking paralogs exhibited a more dramatic response than the assembly factors (Fig. EV3B). We extended this rationale to other complexes. We partitioned Complex I into its constituent modules, namely N, Q, and P, and monitored the progression of each individually. The more aggressive phenotype was observed in genes belonging to the proton translocation module (P module), which is involved in proton pumping, compared to the N module and Q-module, which are involved in NADH oxidation or ubiquinone (Q) reduction, respectively (Kravchuk et al, 2022) (Fig. EV3C). This suggests that the generation of the electrochemical proton gradient through CI proton pumping is more significant than its NADH dehydrogenase activity in sensitizing cells to galactose conditions. In addition to its role in proton translocation, the P module may also contribute to the biogenesis or proper assembly of other respiratory chain complexes, such as CIII and CIV, which could explain the more severe phenotype associated with its disruption (Moreno-Lastres et al, 2012). This broader impact highlights the potential of the P module to influence overall respiratory chain integrity and mitochondrial function. For CII, we observed a consistent phenotype among its subunits, except for SDHA, which showed no phenotype (Fig. EV3D). Lastly, CV (or ATP synthase) was dissected into its major structural components, referred to as F1 and F0, and we tracked their phenotypic progression over time. Here, genes located within the F1 region, responsible for catalyzing the synthesis of ATP, exhibited a more pronounced phenotype (Fig. EV3E).

Our analysis also depicted a progressive increase in the detection of assembly factors showing positive scores across most complexes as time elapsed. This phenomenon was particularly evident in the context of CI, where the identification of assembly factors increased nearly threefold between 24 h and 1 week (Fig. 3A). In the case of

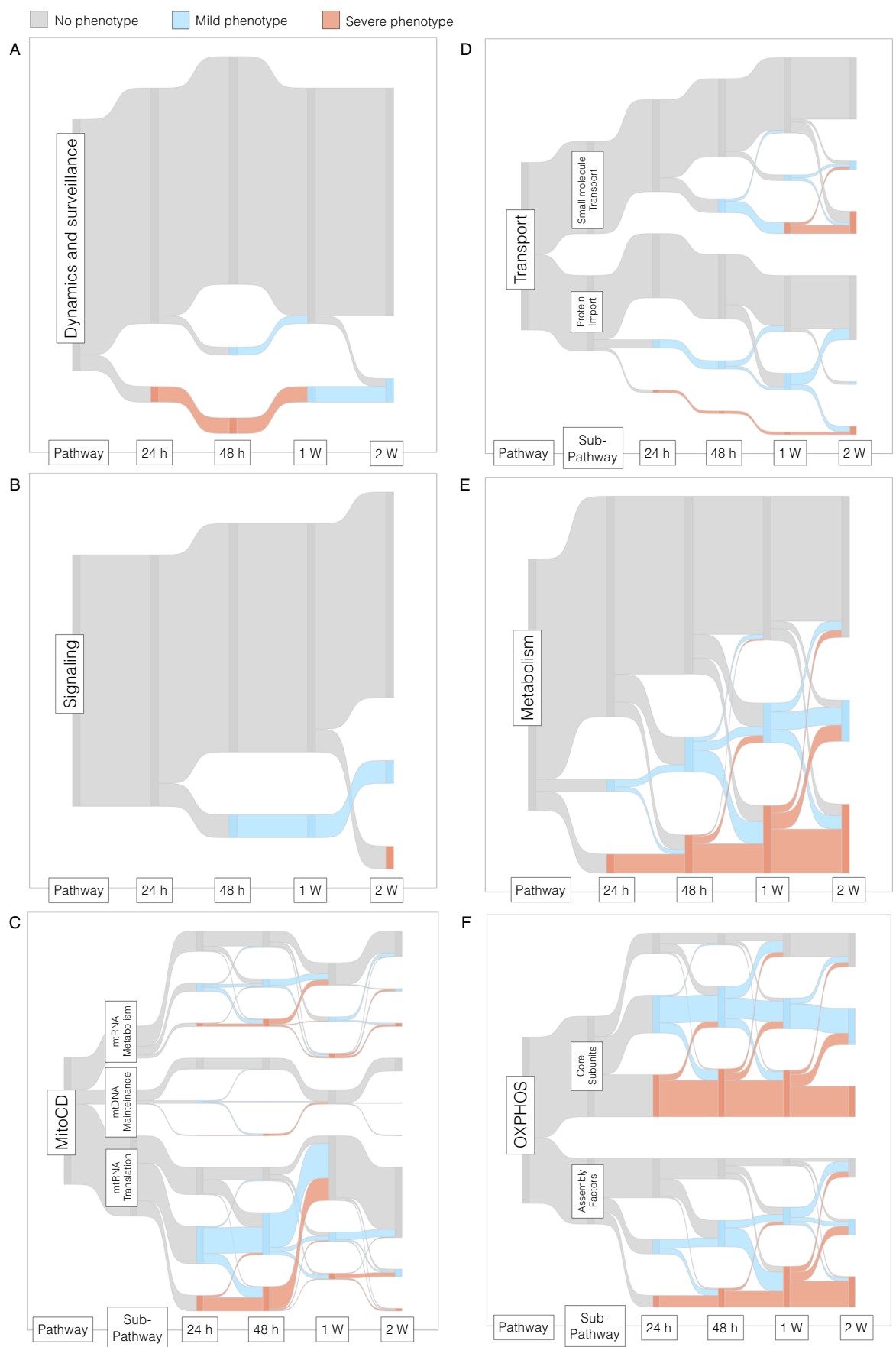

**Figure 2. Temporal mapping of key mitochondrial pathways involved in oxidative metabolism.**

Sankey diagrams depicting the progression of genes regarding their phenotype, categorized within the following functional groups: (A) Dynamics and surveillance, (B) Signaling, (C) Mitochondrial central dogma, (D) Transport, (E) Metabolism, and (F) OXPHOS. A drop in the fold change in the gal/glu ratio is considered to indicate no phenotype when below 20%, a mild phenotype between 20 and 35%, and a severe phenotype above 35%. Source data are available online for this figure.

CIII, we observed a steady linear escalation in assembly factors, peaking at 2 weeks, while for CV the maximum stabilizes at 1 week (Fig. 3A). Overall, this underscores the importance of incorporating multiple time points in this type of screenings facilitating the identification of a growing number of assembly factors and potential regulators of OXPHOS, including those with subtle phenotypes.

## Identification of uncharacterized assembly factors and regulatory components of mitochondrial oxidative metabolism

We leveraged combined MAGeCK (Li et al, 2014) and clustering (*clust*) computational analysis (Abu-Jamous and Kelly, 2018) to identify novel, uncharacterized genes strongly correlated with the temporal dynamics of OXPHOS components. Raw data from both cell lines, encompassing various time points, were utilized to generate a ranked dataset comprising average phenotypic values at the gene level via MAGeCK. Subsequently, a clustering method using parameter sweeping was employed. This process identified a total of 23,933 phenotypic-based clusters. These clusters were evaluated through a hypergeometric distribution for their enrichment of genes associated with OXPHOS, both individually and collectively, encompassing core components and assembly factors. Finally, using the clusters with the best $p$ value ($p < 0.00001$) we compiled a curated list comprising the most frequent gene candidates (Dataset EV2), which refers to genes found within the clusters but not functionally assigned as OXPHOS components (Fig. 3B). From this list, we directed our focus towards genes not previously linked to OXPHOS function. Based on these criteria, we selected RTN4IP1 and ECHS1 for follow-up validation (Fig. 3C,D). RTN4IP1 is classified as an uncharacterized gene in the Mitocarta database (Calvo et al, 2016). Additionally, we found ECHS1 to be an intriguing candidate because it is the only gene involved in fatty acid oxidation (FAO) whose mutations in patients manifest clinical characteristics compatible with Leigh syndrome (LS), a disease commonly associated with OXPHOS defects.

## RTN4IP1 depletion selectively impairs Complex I levels and activity

Among the list of candidates potentially implicated in OXPHOS activity, we further investigated reticulon 4-interacting protein 1 (RTN4IP1), also known as optic atrophy-10 (OPA10), due to its status as a poorly characterized gene lacking clear functional annotation (Angebault et al, 2015). RTN4IP1 has recently been reported to function as an NADPH oxidoreductase involved in coenzyme Q (CoQ) biosynthesis (Park et al, 2024). However, whether RTN4IP1 is essential for proper OXPHOS function and the molecular mechanism underlying the pathological phenotype observed in patients is unknown. We confirmed that RTN4IP1-null cells were unable to thrive under galactose and perished under these conditions

(Fig. 4A). This phenotype was likely attributed to a disruption in OXPHOS functionality as whole-cell oxygen consumption was almost undetectable in RTN4IP1 KO cells (Fig. 4B). Furthermore, mitochondria from RTN4IP1 KO cells showed minimal oxygen consumption when driven by pyruvate and malate (Complex I substrates), while oxygen consumption driven by succinate (a Complex II substrate) was unaltered (Fig. 4C). These phenotypes were rescued by ectopically expressing RTN4IP1 in KO cells (Fig. EV4A–C). To elucidate the role of RTN4IP1, we utilized a previously established methodology to analyze the Achilles dataset via the cancer dependency map portal (DepMap) (Pacini et al, 2024), aiming to identify genes that exhibit coessentiality with RTN4IP1. This analysis revealed that among the top 50 genes co-essential with RTN4IP1, a significant portion are implicated in essential cellular processes, including ETC function, mitochondrial translation/transcription, and CoQ biosynthesis. Notably, the most enriched genes were associated with mitochondrial complex I (Fig. 4D). Consistent with this observation, blue native analysis performed on digitonin-solubilized mitochondria revealed a total absence of mature forms and activity of CI super assembly with other complexes in RTN4IP1 KO cells (Fig. 4E,F). This defect was rescued by ectopic overexpression of RTN4IP1 in these cells (Fig. EV4D). To gain further insights into how RTN4IP1 disruption could impact cell biology, we conducted proteomics analysis on whole-cell samples from RTN4IP1 WT and KO cells. Among the approximately 4800 proteins identified, we observed an upregulation in 129, while 149 were significantly downregulated (Fig. EV4E; Dataset EV3). Pathway analysis revealed that the affected biological processes of upregulated proteins were related to cell-substrate adhesion and extracellular organization. Interestingly, downregulated proteins were primarily involved in the mitochondrial electron transport chain, NADH dehydrogenase complex assembly, and mitochondrial respiratory chain complex I (Fig. EV4F). Proteins downregulated in the OXPHOS system were notably enriched in complex I, whereas the protein subunits of complexes II, III, IV, and V did not exhibit a significant reduction (Fig. 4G). Intriguingly, the sole protein showing significant upregulation in complex I was ACAD9, an assembly factor that, in conjunction with ECSIT and NDUFAF1, is critical for the early steps of CI assembly (Xia et al, 2021). These findings suggest the involvement of RTN4IP1 in the assembly of complex I, possibly at the level of the Q-module.

## RTN4IP1 functions as an assembly factor for Complex I involved in the progression of the Q-ND1 module

To gain a comprehensive understanding of the effects of RTN4IP1 depletion on respiratory chain complex assembly and the steady-state levels of other mitochondrial proteins, we employed complex-ome profiling, a quantitative mass spectrometry assay that has previously yielded valuable insights into the investigation of other genes associated with complex I biogenesis (Guerrero-Castillo et al, 2017).

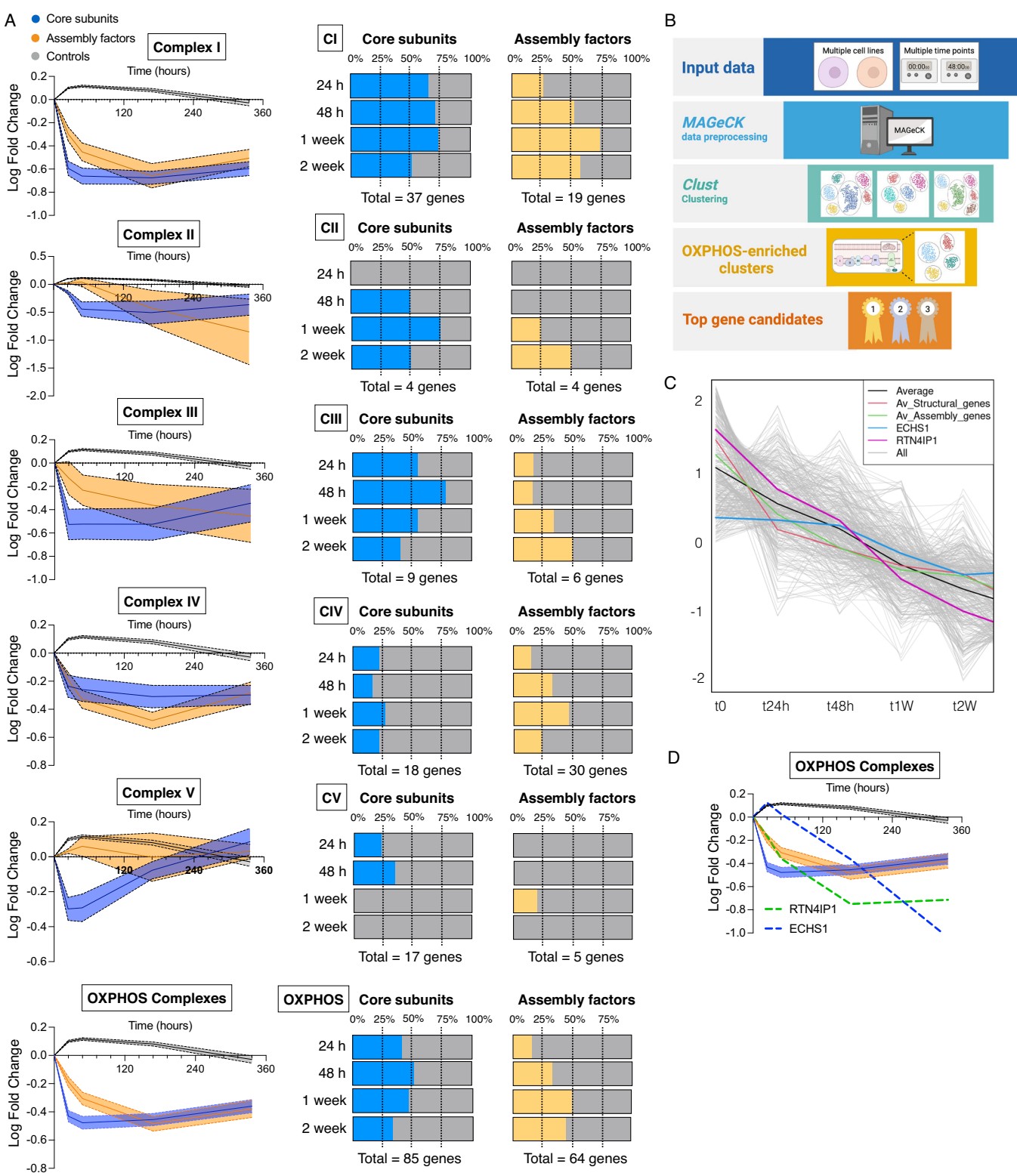

We conducted an in-depth comparison of the BN-PAGE migration profiles using isolated mitochondria from WT and RTN4IP1 KO samples. Eleven distinct bands, corresponding to regions enriched with CI components of the ETC, were sliced and analyzed using mass spectrometry (Fig. EV5A). We identified

41 subunits, accounting for 93% of the mass of CI, and 14 bona fide assembly factors (Dataset EV4). As expected, the detection of peptides corresponding to RTN4IP1 in KO samples was severely limited (Fig. EV5B). When analyzing the different modules involved in CI assembly, we observed significant differences in the higher

**Figure 3. Evolving behavior of OXPHOS components revealed through high-resolution dynamic timeline.**

(A) Temporal evolution of the logarithmic fold change in the gal/glu ratio of OXPHOS complexes (I-V) showing the differences in the dynamic behavior of core subunits and assembly factors. Note that no changes were observed in the control group (sgRNA guides targeting intergenic regions) (Left) and the percentage of genes scoring positive at each time point among all the core subunits or assembly factors for each of the OXPHOS complexes (I-V) (right). (B) Workflow depicting our bioinformatic analysis to identify uncharacterized genes whose function is linked to OXPHOS. (C) An example of one of the clusters that was highly enriched for OXPHOS genes, containing several candidate genes, including ECHS1 and RTN4IP1. (D) A graph that illustrates the phenotypic behavior of OXPHOS complexes and the gene candidates RTN4IP1 and ECHS1. Source data are available online for this figure.

molecular weight bands. RTN4IP1 KO cells showed a substantial reduction in these CI-containing modules, likely at the level of supercomplexes, indicating a notable deficiency in the assembly of the mature form of CI. These differences were particularly pronounced in modules N/Q and Pd-b (Figs. 5A and EV5C). We hypothesized that RTN4IP1 deficiency would lead to an accumulation of the specific module involved in its assembly, while the subsequent modules would experience a decrease in their levels. Indeed, we observed a specific accumulation of the Q-ND1 module in the KO, indicating a halt in the biogenesis progression of this module due to the absence of RTN4IP1 (Fig. 5B). To further test this, we used our complexome profiling data to calculate the ratio between KO and WT cells for all the assembly factors dedicated to each module and submodule in CI biogenesis. TIMMDC1, NDUFAF3, and NDUFAF4, which are critical factors for the assembly of the ND1 module (Sánchez-Caballero et al, 2016), accumulated massively in the KO. Similarly, ACAD9 and, to a lesser extent, DMAC1, which are involved in the assembly of the ND2-Pp-b module (Xia et al, 2021), were also increased in the KO samples. However, other components of this module, such as ECSIT, NDUFAF1, and TMEM126B, showed no alterations (Fig. 5C). The remaining assembly factors involved in the assembly of other modules, mostly at the interface between the Q- and N-module (NDUFAF2) and Pd-a module (TMEM70, DMAC2, and FOXRED1), were markedly reduced in the KO samples (Fig. 5C). These results were further validated by analyzing band 7, which revealed significant differences in these assembly factors between RTN4IP1 KO cells and controls (Fig. EV5D). This suggests that RTN4IP1 is likely to participate in assisting the late stages of CI assembly, specifically the binding of Q-ND1 module to N module, and Pd-b to Pd-a module. Further supporting this hypothesis, we observed that structural components Q-ND1 module, such as NDUFS7, NDUFS3, NDUFS8, NDUFA5, and ND1 itself, were among the subunits that were less reduced in KO samples compared to WT. Conversely, the KO/WT ratio of subunits from other modules that formed later in CI biogenesis, such as those belonging to the Pd-b and N-modules, was more reduced. This reduction was likely because these modules could not fully integrate to form a mature CI and consequently were degraded (Fig. 5D). Blue native analysis of dodecyl-β-D-maltoside (DDM)-solubilized mitochondria confirmed the accumulation of Q-ND1 module subassemblies. However, no accumulation was observed for the N-module or Pd-b module, as determined using specific antibodies against their respective subunits (Fig. EV5E). Collectively, these findings provide compelling evidence that RTN4IP1 is a critical factor in promoting the late-stage assembly of CI.

Since RTN4IP1 has been implicated in CoQ biosynthesis by assisting the O-methyltransferase function of COQ3 (Park et al,

2024), we sought to determine if reduced CoQ levels were indeed the cause of the observed decrease in mitochondrial respiration. First, we confirmed that RTN4IP1 cells had reduced levels of $CoQ_{10}$ and $CoQ_9$, while the intermediate of the CoQ pathway, $DMQ_{10}$, accumulated (Fig. EV5F). Next, we supplemented RTN4IP1 KO cells with $CoQ_{10}$ for a week, and CI levels were assessed. Blue Native analysis did not show a restoration of CI levels in CoQ-treated cells, suggesting that the function of RTN4IP1 in CoQ biosynthesis is distinct from its role in CI assembly (Figs. 5E and EV5G). Additionally, *COQ7* mutant cells, characterized by diminished CoQ levels due to defective CoQ synthesis (Wang et al, 2022), exhibited comparable CI levels to those of control cells (Fig. EV5H), suggesting that the reduced CoQ levels are not responsible for the decreased CI levels. In summary, these findings suggest that RTN4IP1 not only regulates CoQ biosynthesis but also plays a crucial role as an assembly factor for CI, specifically within the Q-module responsible for CoQ binding (Fig. 5F).

## Disrupting ECHS1 leads to the accumulation of valine catabolism intermediates and impairs lipoic acid-dependent enzymes

Unlike the other FAO enzymes, mutations in ECHS1 result in clinical presentations characterized by Leigh syndrome (Sun et al, 2020). This pathology is believed to be caused by combined respiratory chain deficiency secondary to ECHS1 disruption (Sakai et al, 2015), although the molecular mechanism linking these processes remains unknown. Indeed, we confirmed that ECHS1 was the only enzyme involved in FAO that scored positive in our screening data (Fig. 6A). ECHS1 KO cells were unable to sustain long-term growth in galactose (Fig. 6B) and displayed reduced mitochondrial respiration (Fig. 6C). Of note, cell lacking ACADS1, which is involved in an upstream step in FAO, did show normal oxygen consumption levels. Ectopic expression of ECHS1 rescued these phenotypes (Fig. EV6A–C). To determine whether impaired oxygen consumption was due to defects in the levels of the respiratory OXPHOS complexes, we performed Blue Native analysis in ECHS1 WT and KO digitonin-solubilized mitochondria. We detected a subtle decrease at the level of supercomplexes without affectation of the isolated forms of complexes CIII, CIV and CII (Fig. 6D). Nonetheless, these marginal changes are unlikely to explain the observed decrease in respiration and galactose-mediated cell death in ECHS1 knockout cells. We were puzzled by the specific phenotype of ECHS1 compared to the rest of the FAO enzymes and sought to corroborate whether inhibiting FAO could yield similar phenotypes. Treatment with etomoxir, which inhibits FAO by preventing the import of long-chain fatty acids into the

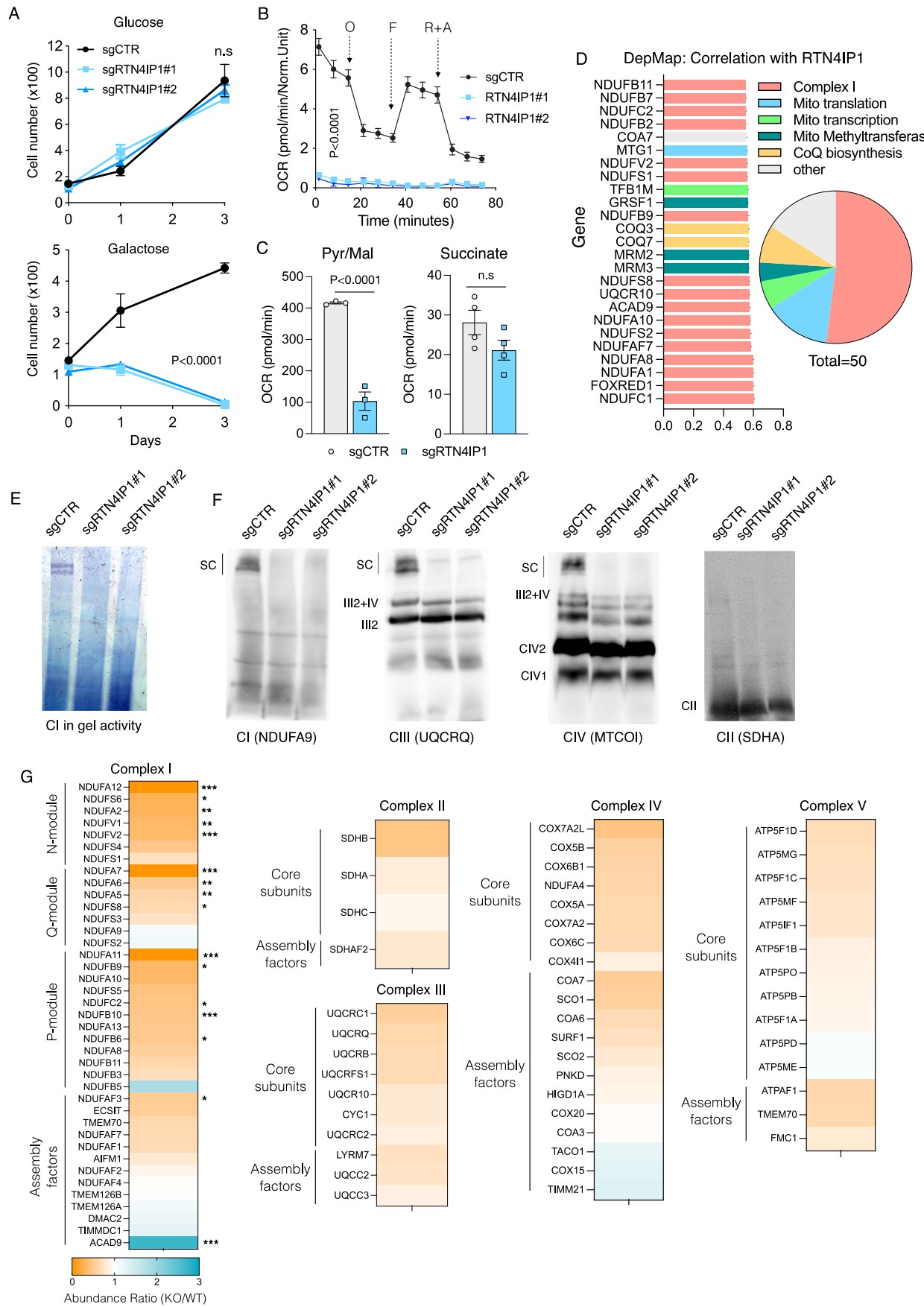

**Figure 4. RTN4IP1 depletion leads to specific reductions in Complex I levels and activity.**

(A) Growth curve of control and two stable clones of RTN4IP1 KO HeLa cell cultures in either glucose or galactose for the indicated days. (B, C) Seahorse analysis showing the oxygen consumption rates (OCR) in intact cells of control and two stable clones of RTN4IP1 KO HeLa cells, or in isolated mitochondria, using pyruvate-malate or succinate as specific substrates, respectively (C). (D) RTN4IP1 demonstrates coessentiality with mitochondrial complex I genes. The GO annotation reveals the top 25 (left) and 50 (right) genes most highly correlated with RTN4IP1. Mito mitochondrial. (E, F) Complex I in gel activity (E) and immunodetection of the indicated proteins representing CI, CIII, CIV, and CII after blue native PAGE (BN-PAGE) of digitonin-solubilized mitochondria from control and two stable clones of RTN4IP1 KO HeLa cells (F). (G) Proteomic heatmap illustrating differences in the abundance of complex I subunits between control and RTN4IP1 KO cells. Proteomic heat-map analysis depicting the relative enrichment of identified proteins for Complex I, II, III, IV, and V in control and RTN4IP1 KO HeLa cells. Immunoblots shown are representative of >3 independent experiments, growth curves are calculated from three replicates, and OCR is calculated from at least three replicates. Data were presented as means ± SEM. The statistical significance of the differences between groups was determined by paired two-tailed Student's *t*-test (A, B, G) and two-way ANOVA (G). n.s not significant. O oligomycin, F FCCP, R + A rotenone plus antimycin a. Pur/Mal pyruvate/malate. Source data are available online for this figure.

mitochondria, did not alter cell proliferation or survival of galactose-grown cells (Fig. EV6D). This suggests that FAO does not account for the phenotype observed in the ECHS1 cells. As ECHS1 may also participate in the catabolism of branched-chain amino acids (BCAAs) (Ferdinandusse et al, 2015), we examined our screening datasets for enzymes involved in BCAA metabolism that exhibit a similar phenotype to ECHS1. Interestingly, HIBCH, a 3-Hydroxyisobutyryl-CoA Hydrolase that lies downstream of ECHS1 in valine catabolism, also yielded positive results (Fig. EV6E). Furthermore, HIBCH KO cells also exhibited reduced oxygen consumption and a proliferative defect when grown under galactose conditions (Fig. EV6F,G). This suggests that disrupting valine metabolism, particularly at these two steps, can compromise mitochondrial oxidative metabolism, sensitizing cells to galactose-induced cell death. We reasoned that if disrupting BCAA metabolism compromised mitochondrial oxidative capacity somehow, then limiting access to the BCAAs valine, leucine, and isoleucine should ameliorate the phenotype. Consistent with this hypothesis, seahorse analysis revealed that ECHS1 KO cells cultured in a medium lacking BCAAs exhibited elevated levels of oxygen consumption (Fig. 6E). Next, we dissected the contribution of each specific amino acid by culturing ECHS1 KO cells in media without either valine, leucine, or isoleucine. Interestingly, ECHS1 KO cells cultured without valine showed a significant restoration in their oxygen consumption levels compared to those same cells cultured without leucine or isoleucine and were able to sustain growth under galactose conditions (Figs. 6F and EV6H). This indicates that disrupting valine metabolism by ablating ECHS1 or HIBCH causes a deficit in oxidative metabolism. Notably, supplementation of ECHS1 KO cells with high doses of valine was enough to further decrease their mitochondrial respiration (Fig. EV6I). To further strengthen these results, we restricted BCAAs utilization proximal to ECHS1 by deleting the ACAD8 gene, which acts upstream of ECHS1. ECHS1/ACAD8 double KO cells showed a similar restoration in their oxygen consumption compared to ECHS1 KO cells deprived of BCAAs (Fig. 6G). One potential shared mechanism underlying the specific phenotype observed exclusively in ECHS1 and HIBCH KO cells is the accumulation of upstream metabolites methacrylyl-CoA and 3-hydroxyisobutyryl-CoA. These metabolites, characterized by their high reactivity, have the potential to form adducts through interaction with sulfhydryl groups (Haack et al, 2015). We leveraged our screening datasets to identify top-scoring genes and pathways in which thiol groups play an essential role in their function. Lipoic acid metabolism emerged as the top-ranked pathway in our analysis, correlating with ECHS1

at later time points (Fig. EV6J,K). All the genes involved in the formation of lipoic acid exhibited a pronounced phenotype under galactose conditions (Fig. EV6L). Lipoic acid serves as a cofactor for numerous mitochondrial enzymes, including pyruvate dehydrogenase, α-ketoglutarate dehydrogenase, and branched-chain ketoacid dehydrogenase (Solmonson and DeBerardinis, 2018). In our screening, genes associated with the pyruvate dehydrogenase complex (PDHX, PDHB, DLAT, DLD, and PDHA1) exhibited similar scores to ECHS1, whereas genes related to branched-chain ketoacid dehydrogenase (BCKDHA and BCKDHB) did not display any notable phenotype (Fig. EV6M). In this regard, pyruvate dehydrogenase activity was reduced in the ECHS1 KO cells, and it was restored in ECHS1/ACAD8 double KO cells (Fig. 6H). This correlated with decreased levels of total protein lipoylation and reduced lipoylation of the pyruvate dehydrogenase complex subunit E2 (dihydrolipoyl transacetylase) in mitochondrial extracts of ECHS1 KO cells, which was rescued in ECHS1/ACAD8 double KO cells (Figs. 6I and EV6N). Further experiments confirmed that immunoprecipitated pyruvate dehydrogenase complex subunit E2 showed reduced levels of lipoylation in ECHS1 KO cells, which were again normalized in double KO cells (Fig. 6J).

Altogether, these results unveil a mechanism whereby the disruption of branched-chain amino acid metabolism, specifically at the enzymatic steps involving ECHS1 and HIBCH, impairs mitochondrial pathways reliant on sulfur metabolism. This impairment leads to decreased levels of protein lipoylation, particularly affecting enzymes such as pyruvate dehydrogenase and, likely, OGDH, which ultimately compromises oxidative metabolism and reduces cellular fitness (Fig. 6K).

## Discussion

Enhanced comprehension of mitochondrial genetics has bolstered diagnostic capabilities in recent years. Nonetheless, a significant subset of individuals diagnosed with mitochondrial diseases based on biochemical markers do not harbor mutations in established mitochondrial disease genes. This implies the existence of unidentified proteins pivotal for OXPHOS function, whose pathogenic potential could be assessed through a galactose-specific lethality phenotype. In this study, we aimed to achieve a comprehensive understanding of the regulatory factors governing oxidative metabolism and their potential implications in human disorders. To accomplish this, we devised and executed a time-resolved CRISPR/Cas9 screening strategy exclusively targeting

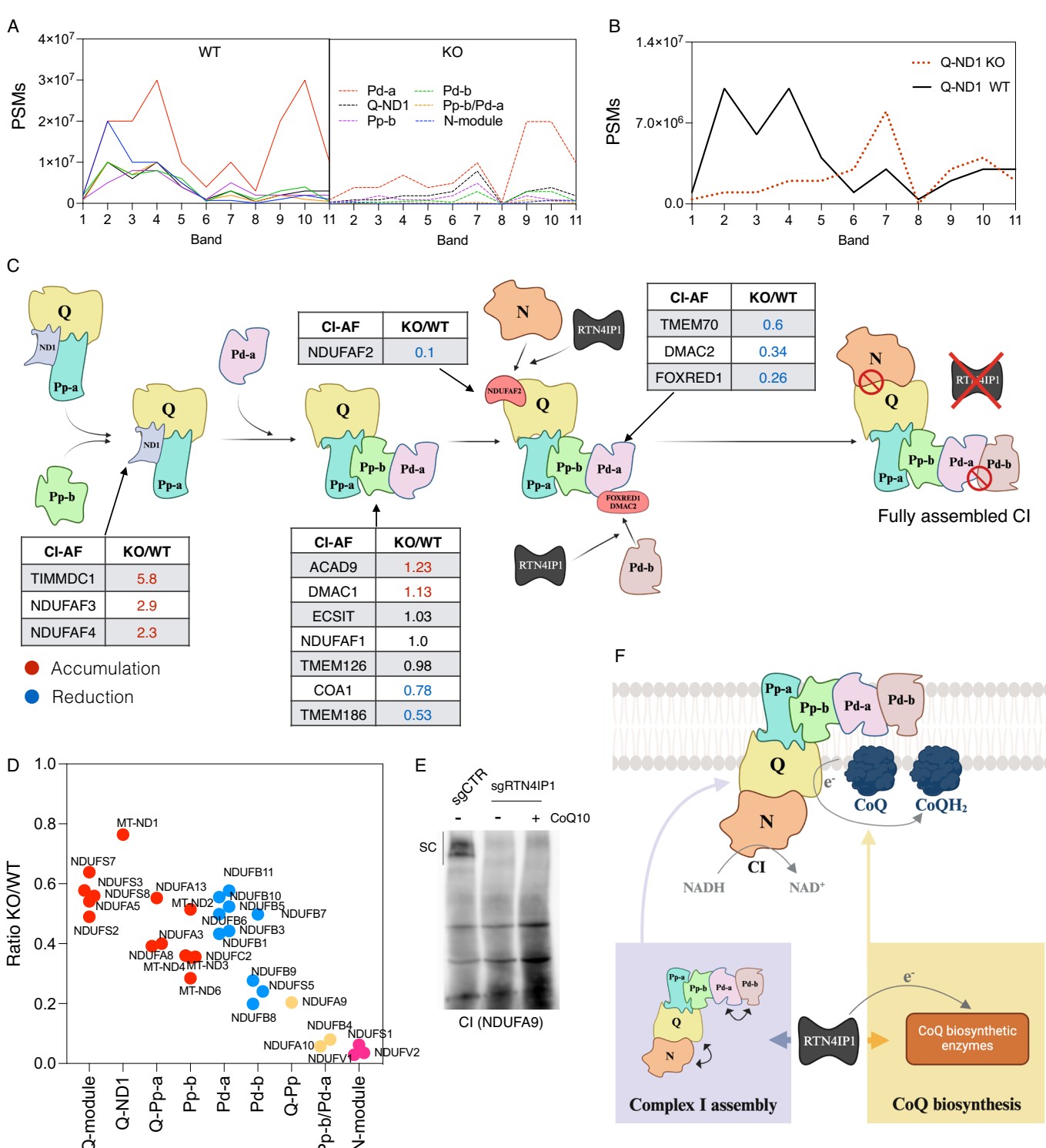

mitochondrial genes, which yielded a robust inventory of genes essential for mitochondrial oxidative metabolism. Unlike previous studies, our investigation revealed the temporal dynamics of these changes, enabling the identification of previously uncharacterized genes that play pivotal roles in regulating the function and integrity of OXPHOS complexes. A notable example of this is the

identification of RTN4IP1 as a new assembly factor involved in the proper maturation of CI. Recessive mutations in RTN4IP1 have been identified through whole-exome sequencing in several patients exhibiting severe central nervous system diseases and optic atrophy (Angebault et al, 2015). However, the specific mechanism by which RTN4IP1 operates has remained obscure owing to its

**Figure 5.   RTN4IP1 is a critical assembly factor for the proper progression of the Q-ND1 module in mammalian Complex I.**

(A) Quantitative profiles of the indicated CI modules between WT and RTN4IP1 KO, expressed in total peptide-spectrum matches (PSMs). (B) Quantitative profiles of the Q-ND1 module between WT and RTN4IP1 KO, expressed in total peptide-spectrum matches (PSMs). (C) Proposed model of CI assembly showing the KO/WT ratio for each of the assembly factors (AF) involved in modules detected in our proteomic analysis. In red are the AF that accumulate in the KO samples, while in blue are those that decrease. We also illustrate where RTN4IP1 participates according to our observations. (D) Analysis of the KO/WT ratio demonstrates that structural components within the Q-module and Q-ND1 module, where RTN4IP1 is potentially implicated, exhibit lower reduction compared to other modules. (E) Restoration of CoQ10 levels in RTN4IP1 KO HeLa cells did not rescue defective CI assembly assessed by Blue native PAGE (BN-PAGE). (F) Model contextualizing the dual role of RTN4IP1, highlighting its involvement in CoQ biosynthesis and its participation in the progression of the Q-ND1 module of Complex I (CI), which is responsible for the reduction of CoQ. Immunoblots shown are representative of >3 independent experiments. PSMs peptide-spectrum matches. Source data are available online for this figure.

uncharacterized nature. Recent studies have highlighted the role of RTN4IP1 as an NADPH oxidoreductase participating in the biosynthesis of coenzyme Q (CoQ) (Park et al, 2024). While we confirmed the involvement of RTN4IP1 in regulating intracellular levels of CoQ, our results strongly suggest that this function is distinct from its role in controlling CI assembly. Proposing that this decoupling is likely the underlying cause of the severe neurodegenerative phenotype observed in patients, it would be intriguing to elucidate the specific phenotypes arising from defective CI assembly and those more specific to CoQ defects. The discovery of a factor capable of simultaneously regulating CI assembly and CoQ biosynthesis is particularly surprising and highlights a previously unrecognized level of coordination between these two crucial processes. Whether RTN4IP1 functions through interactions with other partners or by facilitating posttranslational modifications during CI assembly remains unanswered and warrants further investigation.

By leveraging the temporal dynamics of our screening, we delineated the profound impact of ECHS1 loss-of-function on mitochondrial oxidative metabolism. Typically, patients with deficiencies in this FAO enzyme are diagnosed with Leigh syndrome (Sun et al, 2020), a lethal form of subacute necrotizing encephalomyelopathy characterized by OXPHOS dysfunction. However, the molecular basis of this association has remained elusive. In this study, we present compelling evidence indicating that this phenotype is not contingent on FAO but rather on disrupted BCAA metabolism. Our model implicates the accumulation of toxic, highly reactive intermediate metabolites (methacrylyl-CoA and 3-hydroxyisobutyryl-CoA), which can spontaneously react with sulfhydryl groups and form adducts with lipoic acid. This process negatively affects enzymes like the PDH complex, which depend on this cofactor for function. Analysis of metabolites in the urine of patients with ECHS1 mutations corroborated our model, showing elevated levels of cysteine/cysteamine conjugates originating from valine metabolites (Ferdinandusse et al, 2015). Intriguingly, subtle or no accumulation of lipid degradation metabolites, such as acylcarnitines or free fatty acids, was observed, which stands in stark contrast to the significant accumulation of these metabolites in patients with mutations in other FAO enzymes. An intriguing phenomenon observed in some ECHS1-deficient patients is the occurrence of secondary OXPHOS defects (Burgin et al, 2023; Sakai et al, 2015). In this regard, we also observed a small reduction in some of the OXPHOS complexes, particularly in CI-bearing supercomplexes. While various hypotheses have been proposed, including disruptions in OXPHOS protein complex

biogenesis or stability, the precise mechanism underlying this effect remains elusive. Building on the evidence presented in our study, we propose that the accumulation of reactive intermediates from BCAA metabolism may interact and interfere with iron-sulfur clusters essential for the formation of certain OXPHOS complexes. Notably, a recent study showcased clinical enhancements subsequent to the implementation of a low-valine diet in a pediatric patient diagnosed with ECHS1 deficiency (Pata et al, 2022). Our work offers a mechanistic explanation for this observation and suggests that inhibiting upstream enzymes involved in branched-chain amino acid metabolism, such as ACAD8, may benefit these patients. A recent study also presented a double knockout of ECHS1/ACAD8 (Houten et al, 2023), but in their case, this intervention did not rescue the lipoylation of PDH complex components on its own. This highlights a potential tissue or cell line specificity in these processes, suggesting that the response to such genetic alterations may vary across different experimental conditions.

An evident advantage of our small, mitochondria-focused library, in contrast to conventional genome-wide libraries utilized in previous studies (Arroyo et al, 2016), lies in its ability to detect significant changes despite subtle phenotypes. ECHS1 and HIBCH serve as clear examples of this phenomenon, explaining why these genes may have eluded detection in prior analyses. This holds particular clinical relevance, given that mutations in these genes underlie various human disorders. The incorporation of multiple time points into our analysis provided our approach with sufficient resolution to discern genes that, despite showing minor phenotypic alterations, yield notably positive scores with high statistical significance. Such capability is exemplified by the capture of genes such as PYURF, C5orf63, IBA57, SUCLG1, and MRRF. Another example is the ability to discriminate between structural genes with or without paralogues. Importantly, even these subtle phenotypic differences carry clinical significance, as mutations in several of these genes have been associated with human diseases, thus warranting further in-depth investigations.

While our results have confirmed previous findings regarding the essential gene repertoire for OXPHOS, they have also unveiled novel constituents and regulators of mitochondrial oxidative metabolism. However, it is important to acknowledge that these findings may be cell-specific and biased towards proliferating cells exhibiting the Warburg effect. Commonly utilized proliferating cell lines in such screenings may not accurately reflect the metabolic profiles of quiescent cells, such as neurons or muscle cells, which often underlie pathologies observed in patients.

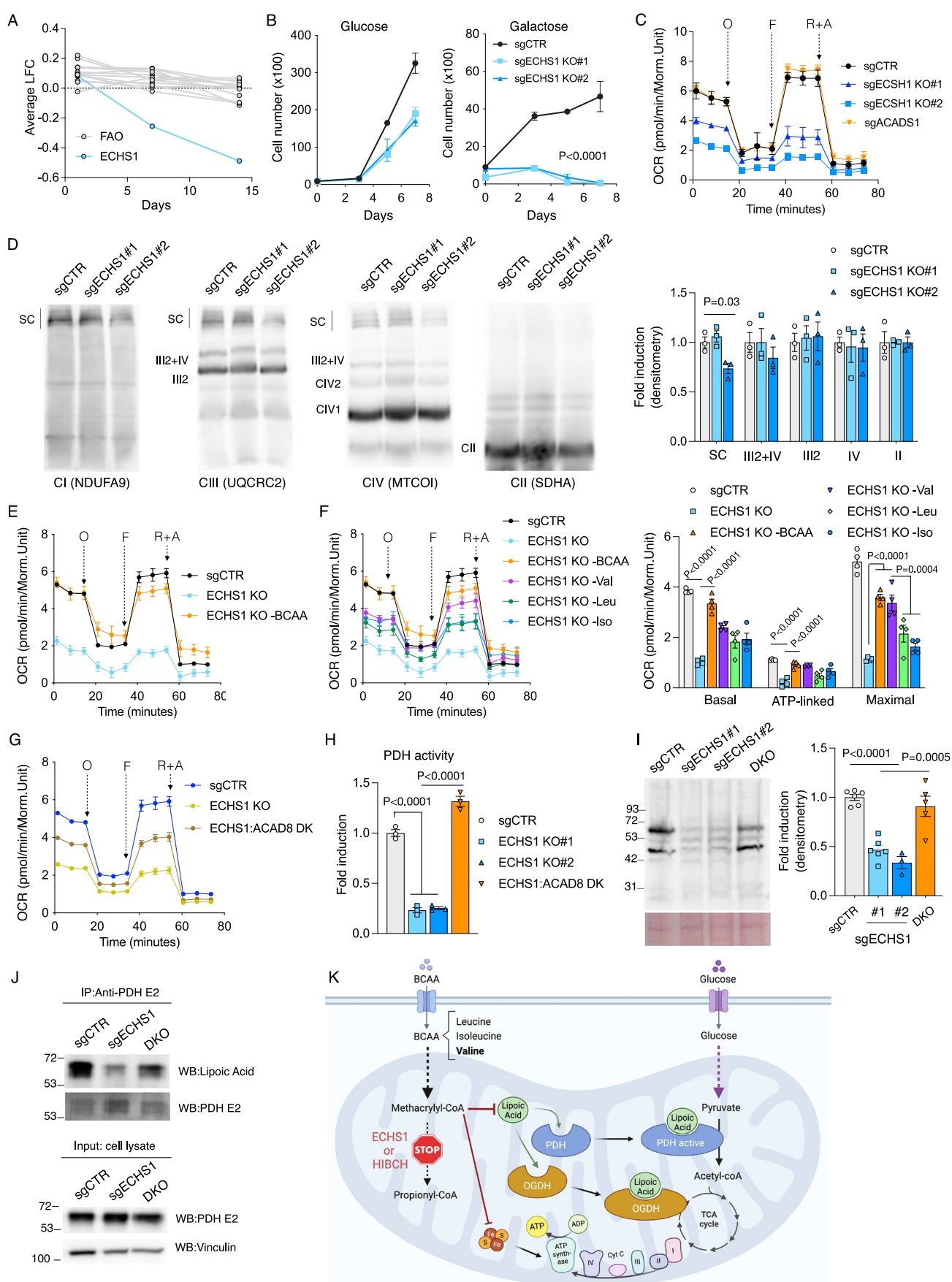

◀ **Figure 6. ECHS1 disruption causes valine catabolism intermediates accumulation and impairs lipoic acid-dependent enzymes.**

(A) Evolution of the phenotype of all fatty acid oxidation-related genes in the screening, along with ECHS1, showing changes over time. (B) Growth curve of control and two stable clones of ECHS1 KO HeLa cells cultured in either glucose or galactose for the indicated days. (C) Seahorse analysis showing the oxygen consumption rates (OCR) of control, two stable clones of ECHS1 KO and ACADS1 KO HeLa cells. (D) Immunodetection of the indicated proteins representing CI, CIII, CIV, and CII after blue native PAGE (BN-PAGE) of digitonin-solubilized mitochondria from control and two stable clones of ECHS1 KO HeLa cells. Quantification of observed changes in protein levels is depicted. (E) Seahorse analysis showing increased respiration in ECHS1 KO cell culture without Branched-chain amino acids (BCAAs). (F) Seahorse analysis (left) and quantification (right) of basal, ATP-linked, and maximal respiration (after rotenone + antimycin subtraction) in control and ECHS1 KO cells cultured without BCAA, valine, leucine, or isoleucine. (G) Seahorse analysis comparing control, ECHS1 KO, and double KO cells for both ECHS1 and ACAD8. (H) Pyruvate dehydrogenase activity in control, two stable clones of ECHS1 KO cells, and double KO HeLa cells for both ECHS1 and ACAD8. (I) Immunodetection (left) and quantification (right) of lipoylated proteins in mitochondrial extracts of control, ECHS1 KO, and double KO cells for both ECHS1 and ACAD8. Ponceau staining was used as a loading control. (J) Immunoprecipitation of the E2 subunit of the PDH complex was performed to assess its lipoylation levels in control, ECHS1 KO, and ECHS1/ACAD8 double KO cells. (K) Proposed model illustrating the disruption of branched-chain amino acid metabolism at the enzymatic steps involving ECHS1 and HIBCH, which impairs mitochondrial pathways dependent on sulfur metabolism. This disruption results in reduced levels of protein lipoylation, notably affecting enzymes such as pyruvate dehydrogenase and OGDH, and potentially the iron-sulfur clusters of the electron transport chain (ETC). Consequently, these impairments compromise oxidative metabolism and decrease cellular fitness. Immunoblots shown are representative of >3 independent experiments, and data were presented as means ± SEM. The statistical significance of the differences between groups was determined by a paired two-tailed Student's $t$-test (B) and two-way ANOVA (D, F, H, I). n.s not significant. O oligomycin, F FCCP, R + A rotenone plus antimycin a. Val valine, Leu leucine. Iso isoleucine, DK or DKO double knockout, PDH pyruvate dehydrogenase complex. OGDH 2-oxoglutarate dehydrogenase complex. Source data are available online for this figure.

# Methods

## Reagents and tools table

| Reagent or resource | Source | Reference |
|---|---|---|
| **Antibodies** | | |
| NDUFA9 | Proteintech | 20312-1-AP |
| NDUFS2 | Abcam | ab192022 |
| ACAD9 | Proteintech | 15770-1-AP |
| SDHA | Invitrogen | 459200 |
| UQCRQ | Invitrogen | PA5-106446 |
| MT-CO1 | Abcam | ab14705 |
| NDUFV1 | Proteintech | 11238-1-AP |
| NDUFB7 | Proteintech | 14912-1-AP |
| lipoic acid | Abcam | AB58724 |
| Vinculin | Abcam | AB207440 |
| Goat Anti-Rabbit Ig | Southern Biotech | 4010-05 |
| Goat Anti-Mouse Ig | Southern Biotech | 1010-05 |
| Goat anti-mouse Dylight 800 | Invitrogen | SA5-10176 |
| Alexa Fluor 680 goat anti-Rabbit | Invitrogen | A21076 |
| **Chemicals** | | |
| DMEM with glucose | Sigma | D5796 |
| DMEM no glucose | Gibco | 11966025 |
| GlutaMAX™ | Thermo Fisher | 61965-059 |
| Fetal bovine serum | Sigma | F0804 |
| MEM non-essential amino acids | Thermo Fisher | 11350912 |
| D(+)-Glucose monohydrate | Merck | 104074 |
| D-(+)-Galactose | Merck | G0750 |
| Sucrose | Merck | S0389 |

| Reagent or resource | Source | Reference |
|---|---|---|
| L-Glutamic acid | Sigma | G1251 |
| Puromycin | Sigma | P8833 |
| Seahorse XF DMEM Medium | Agilent | 103575-100 |
| Oligomycin A | Sigma | 04876 |
| FCCP | Sigma | C2920 |
| Rotenone | Sigma | R8875 |
| Antimycin A | Sigma | A8674 |
| Digitonin | Sigma | D141 |
| Dodecyl-β-D-maltoside | Sigma | D4641 |
| Aminocaproic acid | Sigma | 07260 |
| Imidazole | Sigma | I2399 |
| Blue Comassie G | Serva | 17524.02 |
| Trycine | Sigma | T0377 |
| NADH | Merck | N8129 |
| Nitroblue-tetrazolium | Merck | N6876 |
| (+)-Etomoxir sodium salt hydrate | Merck | E1905 |
| PolyFect | Qiagen | 301105 |
| Polybrene | Merck | TR-1003-G |
| Bradford reagent | Sigma | B6916 |
| Hoechst 33342 | Thermo Fisher | 62249 |
| Triton™ X-100 | Merck | T8787 |
| Protease inhibitor cocktail | Sigma | 11836153001 |
| Laemmli | Bio-Rad | #1610747 |
| 30% Acrylamide/Bis Solution, 37.5:1 | BioRad | #1610159 |
| **Plasmids and virus strains** | | |
| LentiCRISPRV2 lentivirus | Addgene | #52961 |

| Reagent or resource | Source | Reference |
|---|---|---|
| TLCV2 lentivirus | Addgene | #87360 |
| pLX_311-Cas9 expression | Addgene | #96924 |
| pXPR_011-Cas9 activity | Addgene | #59702 |
| psPAX2 lentiviral packaging | Addgene | #12260 |
| pMD2.G lentiviral packaging | Addgene | #12259 |
| 30% Acrylamide/Bis Solution, 37.5:1 | BioRad | #1610159 |
| **Oligonucleotides** | | |
| sgRNA RTN4IP1 Forward | IDT | CACCGGTACTAATCCTTCTAACTGA |
| sgRNA RTN4IP1 Reverse | IDT | AAACTCAGTTAGAAGGATTAGTACC |
| sgRNA ECHS1 Forward | IDT | CACCGCAGCGACTTCCCAACAGCAC |
| sgRNA ECHS1 Reverse | IDT | AAACGTGCTGTTGGGAAGTCGCTGC |
| sgRNA HIBCH Forward | IDT | CACCGTATTAAGAGTCAGTGCATTG |
| sgRNA HIBCH Reverse | IDT | AAACCAATGCACTGACTCTTAATAC |
| sgRNA ACAD8 Forward | IDT | CACCGCAAATTCACCTGGTTCAGTG |
| sgRNA ACAD8 Reverse | IDT | AAACCACTGAACCAGGTGAATTTGC |
| sgRNA ACADS1 Forward | IDT | CACCGACACACCATCTACCAGTCTG |
| sgRNA ACADS1 Reverse | IDT | AAACCAGACTGGTAGATGGTGTGTC |
| **Commercial assays** | | |
| Blood & Cell Culture DNA Midi Kit | Qiagen | 13343 |
| BCA assay kit | Pierce | 23228 |
| Pyruvate Dehydrogenase kit | Abcam | ab287837 |
| Seahorse XFe96/XF Pro FluxPak | Agilent | 103792-100 |
| **Other** | | |
| Heidolph homogenizer Hei-TORQUE 100 | Heidolph | 501-61020-00 |
| Reverse-phase C18 3.5 mm, 4.6 × 150 mm column | Waters | 186005270 |
| Protein G-sepharose beads | Merck | GE17-0618-01 |

## Cell culture

The screening was performed on either human HeLa and 143B cell lines, expressing the spCas9 protein. Subsequent experiments for studying the candidate proteins were performed on human HeLa

cells. Lentiviruses were generated using 293T-X cells (Takara Bio, USA). Human skin fibroblasts were obtained from a healthy donor or a patient with a pathogenic variant in *COQ7* gene, harboring a frameshift deletion (c.161_161delG, p.Val55fs) in exon 2 and a missense mutation (c.319 C > T, p.Arg107Trp) in exon 3.

Cell lines were cultured in media consisting of Dulbecco's modified Eagle's medium (DMEM), 2 mM L-glutamine, 10% inactivated fetal bovine serum, and 1% penicillin/streptomycin (P/S). In case of experiments without branched-amino acids, cells were cultured in DMEM with 10% FBS, 2 mM L-glutamine and P/S, and without valine, leucine, or/and isoleucine. Fibroblasts were cultured in high glucose DMEM-GlutaMAX medium supplemented 1% MEM non-essential amino acids, and 1% P/S. All cells were maintained at 37 °C and 5% $CO_2$.

## CRISPR/Cas9 sgRNA library infection

143B and HeLa cells were first infected with the lentiviral vector pLX_311 expressing the endonuclease Cas9 with 8 µg/ml polybrene for 48 h, and then selected with 5 µg/ml blasticidin for 1 week. After selection, to check Cas9 activity, a vial of cells was infected with the lentiviral vector pXPR_011 with 8 µg/ml polybrene for 48 h and then selected with 1 µg/ml puromycin for 4 days. Then, EGFP intensity was measured in Cas9-expressing and control cells by flow cytometry as described (Doench et al, 2014). To titer the optimal cell density and volume of library lentiviruses, a viral tritation was performed. $36 \times 10^6$ 143B or HeLa cells expressing Cas9 were infected with the Mito-library with 8 µg/ml polybrene, to achieve a 30–50% infection efficiency. Cells were infected for 48 h, then the medium was removed and cells were incubated in fresh medium with 1 µg/ml puromycin for 1 week. Then, two vials of $7 \times 10^6$ cells were harvested for the initial time point (t0h) and the rest of cells were separated in two different media: DMEM containing 25 mM glucose or 25 mM galactose as the sole source of sugar. Cells were incubated in both media and two vials of $7 \times 10^6$ cells were harvested at the different time points (t24h, t48h, t1week, and t2weeks) and frozen at −80 °C until DNA extraction.

## Genomic DNA isolation and identification of sgRNA sequences

Genomic DNA was purified from one of the two vials of all screening samples using the Blood & Cell Culture DNA Midi Kit. DNA was resuspended in 100 µl of nuclease-free water. DNA concentrations were measured with a Nanodrop and 10 µg of DNA of each sample were subsequently sequenced. Samples were submitted and sequenced at the Broad Institute using a HiSeq2000 (Illumina) as previously described (Doench et al, 2016), and sgRNA abundance was measured in all time points.

## Data processing and clustering analysis

Counts data from sgRNA sequencing were initially processed using the MAGeCk software (Li et al, 2014), which generated tables capturing gene presence and abundance at each time point. For each gene, the average presence was calculated based on individual sgRNA measurements. For clustering analysis, genes were grouped based on their temporal behavior using the "clust" software, as previously published (Abu-Jamous and Kelly, 2018). The measures

obtained from Hela and 143B cells were considered as replicates to ensure the robustness and reproducibility of the results. The clust algorithm follows several steps to identify optimal gene clusters: Seed Cluster Generation: clust produces a pool of "seed clusters" by applying k-means clustering multiple times to the data with different K values (4, 8, 12, 16, 20). Consensus Clustering: if multiple datasets are used, consensus clusters are calculated using the Binarization of Consensus Partition Matrices (Bi-CoPaM) method, which consolidates clusters across datasets to form consensus clusters. Evaluation of Seed Clusters: Seed clusters are evaluated using the M-N scatter plots technique, which favors larger clusters with low dispersion values and ensures distinct clusters. Selection of Elite Seed Clusters: Elite seed clusters are selected based on their performance in the M-N scatter plots, distinguished by their size and low within-cluster dispersion. Outlier Removal and Final Cluster Identification: The distributions of within-cluster dispersion are analyzed to remove outliers and identify genes that fit within clusters but were missed in previous steps. A range of parameters was explored to identify optimal clustering conditions, including Clustering Method: k-means, Number of Clusters (K values): 4, 8, 12, 16, 20, Tightness Weight: values between 1.5 and 16 were tested, Outliers Threshold (Q3): threshold values ranging from 1.8 to 4 were evaluated, Minimum Cluster Size: clusters were required to include at least 10 genes. Normalization Between Experiments: different types of normalizations were tested, including No normalization, Quantile normalization, Column-wise mean subtraction, and Normalization Per Gene: various methods were assessed, including No normalization, Z-score normalization, Division by the total (sum) of the row, and Linear transformation to the [0, 1] range. Gene Filtering: The option to exclude genes that did not show changes over time was considered. After extensive parameter testing, the best clustering results were achieved with the following settings: Quantile normalization between experiments, Tightness weight of 2.0, and Q3 threshold of 2.5. For enrichment analysis, clusters were evaluated for enrichment in genes known to be structural components or regulators of the mitochondrial respiratory chain. This enrichment was assessed using the hypergeometric distribution, which calculates the probability of observing a given number of genes from a specific category within a cluster by chance. The hypergeometric test parameters include N (Population size): the total number of genes analyzed; K (Number of success states in the population): the total number of genes known to be involved in the mitochondrial respiratory chain; n (Number of draws): the number of genes in the cluster; and k (Number of observed successes): the number of mitochondrial respiratory chain genes in the cluster. The $p$ value obtained from the hypergeometric test indicates the statistical significance of the observed enrichment, with lower $p$ values suggesting a non-random clustering of mitochondrial genes. Candidate Gene Selection: from the clusters showing significant enrichment, candidate genes lacking known functions were selected for further analysis. This approach aimed to identify new genes potentially involved in mitochondrial function and regulation.

## Creation of knock-out cell lines

For the RTN4IP1, HIBCH, ACADS1, and ACAD8 loss of function, CRISPR-Cas9 mediated knock-outs were performed by transducing HeLa cells with LentiCRISPRV2 lentivirus expressing sgRNAs

targeting the gene of interest as previously described (Shalem et al, 2014). For the ECHS1 knockout, a doxycycline-inducible CRISPR-Cas9 system was used by transducing HeLa cells with TLCV2 lentivirus expressing sgRNAs of interest. Lentivirus were formed by transfection with PolyFect reagent and packaging plasmids in 70% confluent 293T-X cells. Transductions were performed by the addition of lentivirus-containing supernatants in the presence of 10 ug/mL polybrene for 24 h. Then the medium was replaced with fresh DMEM, followed by selection with puromycin (1 μg/ml) for 1 week. In the case of the doxycycline-inducible system, cells were treated with 1 μg/ml doxycycline for 5 days to induce Cas9 expression. Finally, a limiting dilution was performed to obtain different stable single-cell clones for the following experiments.

## Cell proliferation

For proliferation assays, equal numbers of cells were seeded in 12-well plates and allowed to proliferate in DMEM containing glucose 25 mM or galactose 25 mM as the main source of sugar. Cells were Hoechst-stained and relative cell numbers counted using a BioTek Cytation 5 Cell Imaging Multimode Reader (Agilent) at the specific time points.

## Mitochondria isolation

Cells were homogenized in medium A (0.32 M sucrose, 1 mM EDTA, and 10 mM Tris-HCl, pH 7.4) at 4 °C, using a Heidolph homogenizer (600 rpm, 30 strokes). The homogenate was centrifuged for 10 min at 600 × $g$ and 4 °C. The supernatant was then centrifuged twice for 10 min at 10,000 × $g$ and 4 °C. The pellet was resuspended in medium A and the concentration was quantified using the Bradford assay. Mitochondria were stored at 80 °C until the following experiments. Mitochondria that were run in a western blot were previously mixed with Laemmli buffer and boiled for 5 min at 95 °C.

## Immunoprecipitation

To immunoprecipitate pyruvate dehydrogenase subunit E2 (DLAT), $4 \times 10^6$ cells were lysed with TBS (150 mM NaCl, 50 mM Tris-HCl, pH 8), 1% Triton X-100, and protease inhibitors for 10 min. Cells were centrifuged for 5 min at max speed and 4 °C, and supernatants were collected. About 30 μl of protein G-sepharose beads per sample were washed with 500 μl PBS and centrifuged for 1 min at 10,000 × $g$ and 4 °C, and beads were resuspended in 200 μl PBS and mixed with 1:50 DLAT antibody for 10 min. Then, cell supernatants were incubated with DLAT antibody-bound beads overnight at 4 °C. Two different negative controls were also performed: a control with DLAT antibody-bound beads without cell lysate; and a control with beads and cell lysate without DLAT antibody. The following day, all samples were washed 6 times with 300 μl lysis buffer and centrifuge for 1 min at 10000 G and 4 °C. Finally, samples were mixed with Laemmli buffer, boiled for 5 min at 95 °C, and immunoblotted in a western blot.

## SDS PAGE (western blot)

Samples were loaded onto a Bis-Tris gel in electrophoresis buffer (25 mM Tris base, 192 mM glycine, 0.1% SDS) and run at 120 V for 1 h. The gel was transferred to a PVDF membrane using the wet

transfer method in transfer buffer (25 mM Tris base, 192 mM glycine, 20% methanol) at 100 V for 1 h at 4 °C. The membrane was blocked with 5% non-fat milk powder in phosphate-buffered saline with 0.1% Tween 20 (PBST) for at least 30 min. Primary antibodies were diluted in blocking buffer and incubated with membranes overnight at 4 °C. Membranes were washed 3 × 10 min with 1x PBST, incubated for 1 h at 25 °C with secondary HRP-conjugated antibodies or secondary fluorescent antibodies, washed again 3 × 10 min with 1x PBST, and developed using ECL in the Amersham IM-680 imager system. The following primary antibodies were used for western blot analysis: anti-lipoic acid (1/1000), anti-DLAT (1/1000), and anti-vinculin (1/1000).

## Blue native gel electrophoresis (BNGE) and in gel activity

BNGE and CI in gel activity was performed as described (Balsa et al, 2019). To solubilize the electron transport chain complexes, digitonin (4 g per g of mitochondrial protein) or 2% dodecyl-β-D-maltoside (DDM) was added in a buffer containing 1.5 M aminocaproic acid and 50 mM bis-tris/HCl (pH 8). Samples were heated at 80 °C for 5′ and centrifuged for 20′ at 4 °C max speed. About 100 μg of protein were loaded and run on 3–13% gradient Blue Native gels. The gradient gel was prepared in 1.5-mm glass plates using a gradient former connected to a peristaltic pump. Blue cathode buffer (50 mM trycine, 7.5 mM imidazole, 0.02% Coomassie blue G, pH 7) was added in the inner chamber to cover the wells, whereas outer chamber was filled with Anode buffer (25 mm imidazole/HCl, pH 7.0), and electrophoresis was run at 150 V for 30′ at 4 °C. Then, Blue cathode buffer was replaced by Cathode buffer (50 mM trycine, 7.5 mM imidazole, pH 7), and electrophoresis was run at 300 V for 1 h at 4 °C. After electrophoresis, the proteins were transferred onto PVDF membranes using the wet transfer method in Tansfer buffer (25 mM Tris base, 192 mM glycine, 20% methanol) at 100 V for 1 h at 4 °C. Finally, membranes were blocked with 5% non-fat milk powder in PBST for 30 min and sequentially probed with specific antibodies in the same manner as western blot. The following primary antibodies were used: NDUFA9 (1/1000), NDUFS2 (1/1000), ACAD9 (1/1000), SDHA (1/1000), UQCRQ (1/1000), NDUFV1 (1/1000), NDUFB7 (1/1000), and MT-CO1(1/1000).

To assess complex I in gel activity, the gel was incubated with 0.14 mM NADH and 1 mg/ml nitro blue tetrazolium in 100 mM Tris-HCl (pH 7.4) at 37 °C until the color developed.

## Seahorse measurement of mitochondrial respiratory capacity

Bioenergetic profiles were measured using Seahorse XF96-Cell Mito Stress Test (Agilent Technologies). Cells were plated in Seahorse XF96-Cell culture Microplates at a density of 15,000 cells per well one day prior to the assay. One hour before the assay, the medium was changed to Agilent Seahorse XF DMEM medium supplemented with 10 mM D-glucose, 1 mM pyruvate, and 2 mM L-glutamine, to a final volume of 180 μM per well. Cells were incubated in a non-CO2 incubator at 37 °C for 1 h before the measurement. For oxygen consumption rate (OCR), the procedure included initial baseline recordings, followed by successive additions of specific metabolic inhibitors: 1 μM oligomycin, 0.75 μM FCCP, and 1 μM rotenone/antimycin. A total of three measurements of OCR and pH were

taken for each drug injection. After seahorse measurements were done, cell nuclei were Hoechst-stained and relative cell numbers for loading control were counted using a BioTek Cytation 5 Cell Imaging Multimode Reader.

In isolated mitochondria: To minimize variability between wells, mitochondria were first diluted 10x in cold 1x MAS (70 mM sucrose, 220 mM mannitol, 10 mM $KH_2PO_4$, 5 mM $MgCl_2$, 2 mM HEPES, 1.0 mM EGTA, and 0.2% (w/v) fatty acid-free BSA, pH 7.2 at 37 °C).

Stock substrates 0.5 M malic acid, 0.5 M pyruvic acid, or 0.5 M succinate and 0.2 mM ADP, were subsequently diluted to the concentration required for plating. Next, while the plate was on ice, 50 uL of mitochondrial suspension (containing 10 μg of mitochondria) was delivered to each well (except for background correction wells). The XF96-Cell culture Microplate was then transferred to a centrifuge equipped with a swinging bucket microplate adapter, and spun at 2000 × g for 20 min at 4 °C. After centrifugation, 130 uL of prewarmed (37 °C) 1x MAS + substrates pyruvate/malate (10 mM/2 mM) or succinate (10 mM) were added to each well. In the case of succinate-driven respiration, 100 uM Rotenone was also added to the MAS buffer. The plate was then transferred to the Seahorse XF96 Analyzer, and the experiment was initiated.

## Proteomics

For the whole-cell proteomic analysis, label-free quantitative proteomics was performed. Proteins were extracted in 300 μl of lysis buffer with protease inhibitor and 0.5 μl of endonucleases. Samples were reduced and alkylated with TCEP and CAA for 30 min at 60 °C, and a total of 100 μg of protein for each sample was digested in S-Trap columns. About 2 μl of the final tryptic digest was quantified by the QBIT method. About 1 μg of tryptic digest from each sample was analyzed by liquid nano-chromatography coupled to mass spectrometry, using a gradient of 120 min, a flow rate of 300 nl/min, and a reverse-phase column of 50 cm length and 75 μm internal diameter. The analysis was performed on a Thermo Ultimate 3000 liquid nano-chromatograph coupled to a Thermo Orbitrap Exploris 240 mass spectrometer, working in DDA mode. The MS1 and MS2 spectra obtained were loaded into the Proteome Discoverer 2.5 analysis suite for identification and quantification of the tryptic peptides in each sample. A database downloaded from UniprotKB containing Homo sapiens sequences as well as a number of typical laboratory contaminants (keratins, albumins, etc) was used as a database.

### Blue-DiS proteomics

For complexome profiling, mitochondria were isolated from cell pellets (60–70 million). Briefly, pellets were resuspended in homogenizing buffer A (0.083 M sucrose, 10 mM MOPS, pH 7.2), incubated 2 min on ice, and homogenized in a tightly fitting glass-teflon homogenizer. Then an equal volume of buffer B (0.25 M sucrose, MOPS 30 mM, pH 7.2) was added. The following steps were performed as previously described. Three wells per condition were loaded each with 200 mg of 4 g/g digitonin-treated mitochondria in a 3–13% hand-casted native gels. After electrophoresis, gels were stained with Coomassie Brilliant Blue R-250 and sliced into 11 bands. Gel bands were subjected to in-gel digestion. After reduction with DTT (10 mM) and alkylation of Cys groups with iodoaceta-mide (50 mM), modified porcine trypsin (Promega) was added at a

final ratio of 1:20 (trypsin-protein). Digestion proceeded overnight at 37 °C in 100 mM ammonium bicarbonate at pH 7.8. Resulting tryptic peptides were loaded and washed on Evotips and separated in an Endurance 15 cm × 150 μm ID, 1.9 μm beads-EV1106 column using an Evosep one HPLC system coupled to an Orbitrap Eclipse Tribrid mass spectrometer (Thermo Fisher, San José, CA, USA) and a 30 SPD preprogrammed gradient.

MS analysis was performed using the data-independent scanning (DiS) method, with some modifications. Briefly, each sample was analyzed in a single chromatographic run covering a mass range from 390 to 1000 m/z. The DiS cycle consisted of 255 sequential HCD MS/MS fragmentation events with 2.5 m/z-windows from 390 to 900 m/z, and with 4 m/z-windows from 900 to 1000 m/z. HCD fragmentation was performed using a 33 normalized collision energy. MS/MS scans were performed using 70 ms injection time, $3 \times 105$ ions AGC target setting, and 17,500 resolution. The whole cycle lasted a maximum of 18 s, depending on ion intensity during chromatography. For protein identification and quantification, all MS/MS spectra were analyzed using DIA-NN 1.8.1. A spectral library was generated (HS_MetOx_1miss_300-1800.predicted.speclib) and used to validate PSMs by filtering at 0.01 FDR. The maximum number of variable modifications was set to 1, and highly heuristic protein grouping was used to reduce the number of protein groups obtained. Fixed-width center of each elution peak was used for quantification, and interference removal from fragment elution curves was disabled.

The mass spectrometry proteomics data have been deposited in the ProteomeXchange Consortium via the PRIDE partner repository with the dataset identifier PXD061050 and 10.6019/PXD061050

### Quantification of CoQ9 and CoQ10 levels

Cell lipid extraction was performed by mixing the cells homogenate with 50 mM SDS and a proportion 2:1 of hexane:ethanol. After vortexing for 2 min in a glass tube, the mix was centrifuged at 2000 × *g* for 5 min (twice). The upper phase was transferred to a glass tube and evaporated with nitrogen gas. The residue was resuspended in 1-propanol. CoQ9 and CoQ10 levels were determined in the resultant extract using a reverse-phase C18 3.5 mm, 4.6 × 150 mm column coupled to electrochemical detection, as previously described (González-García et al, 2022). Mobile phase consisted of methanol, ethanol, 2-propanol, acetic acid (500:500:15:15), and 50 mM sodium acetate at a flow rate of 0.9 ml/min. Then, a standard curve of CoQ9 and CoQ10 was used for a quantitative estimation. Results were normalized by milligram proteins, which were quantified in tissue or cell homogenates using the Bradford assay. The results were expressed in nanograms of CoQ per milligram of protein.

### Graphics and statistics

Statistical analysis was performed with GraphPad Prism (version 9). Data are expressed as mean ± s.e.m. (standard error of the mean). Comparisons for the two groups were calculated using unpaired two-tailed Student's *t*-tests. Comparisons for more than three groups were calculated using one-way ANOVA. Unless otherwise specified, n represents the number of individual biological replicates and is represented in graphs as one dot per sample.

Gene Ontology enrichment analysis was performed with the ShinyGO version v0.741 of the website of South Dakota State University. RTN4IP1 top 100 codependencies for CRISPR were done with the DepMap Public database on the website https://depmap.org/portal/. Sankey diagrams were plotted with the package *ggsankey* in R version 4.4.0. Graphical figures were designed with Biorender.

## Data availability

Data from the CRISPR screenings are available in datasets EV1 and EV2. Proteomic datasets are available in EV3 and EV4 and have been deposited via ProteomeXchange under the identifier PXD061050. Additional datasets used and/or analyzed in this study are available from the corresponding author upon reasonable request.

The source data of this paper are collected in the following database record: biostudies:S-SCDT-10_1038-S44319-025-00459-9.

## Peer review information

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

## Acknowledgements

We thank all the members of the Balsa laboratory for discussions regarding this project. Dr. Luis C López (University of Granada) kindly provided fibroblasts from a healthy donor and a *COQ7*-mutated patient. Dra. Cristina Ugalde (Fundación de Investigación Hospital 12 de Octubre, imas12) generously provided 143B and HeLa cells. This work was supported by grants to Dr. Balsa from the Spanish Government, Ministerio de Ciencia e Innovación (PID2019-110766GA-I00 and PID2022-137404OB-I00), European Research Council (ERC) under the European Union's Horizon 2020 research and innovation (ERC-2020-STG grant agreement n° 948478), and Fundación CRIS contra el cancer (PR_EX_2022-01). The CBM is supported by Consejo Superior de Investigaciones Científicas and Universidad Autónoma de Madrid, and is a Severo Ochoa Center of Excellence (grant CEX2021-001154-S) funded by MICIN/AEI/10.13039/501100011033. MZD was supported in part by an FPI-UAM PhD fellowship. SLM was supported in part by a Margarita Salas-UAM postdoctoral fellowship (CA4/RSUE/2022-00037). JAE is supported by PID2021-1279880B funded by MICINN/AEI/10.13039/501100011033 and the European Union "NextGenerationEU"/Plan de Recuperación Transformación y Resiliencia -PRTR; "la Caixa" Banking Foundation (project LCF/PR/H23/52430010) and CIBERFES (CB16/10/00282) funded by ISCIII. PFM is supported by FPU21/06416 by a fellowship. The CNIC is supported by the Instituto de Salud Carlos III (ISCIII), the Ministerio de Ciencia e Innovación (MICINN), and the Pro CNIC Foundation, and is a Severo Ochoa Center of Excellence (grant CEX2020-001041-S funded by MICIN/AEI/10.13039/501100011033).

## Author contributions

**Marcos Zamora-Dorta**: Formal analysis; Investigation; Visualization; Methodology; Writing—original draft; Writing—review and editing. **Sara Laine-Menéndez**: Formal analysis; Investigation; Visualization; Methodology; Writing—original draft; Writing—review and editing. **David Abia**: Resources; Software; Validation; Methodology. **Pilar González-García**: Resources; Methodology. **Luis C López**: Resources; Methodology. **Paula Fernández-Montes**: Resources; Methodology. **Enrique Calvo**: Resources; Methodology. **Jesús Vázquez**: Resources; Methodology. **José Antonio Enríquez**: Resources; Visualization; Methodology; Writing—review and editing. **Eduardo Balsa**: Conceptualization; Data curation; Supervision; Funding acquisition; Validation; Investigation; Visualization; Writing—original draft; Project administration; Writing—review and editing.

Source data underlying figure panels in this paper may have individual authorship assigned. Where available, figure panel/source data authorship is listed in the following database record: biostudies:S-SCDT-10_1038-S44319-025-00459-9.

## Disclosure and competing interests statement

The authors declare no competing interests.

# Expanded View Figures

A

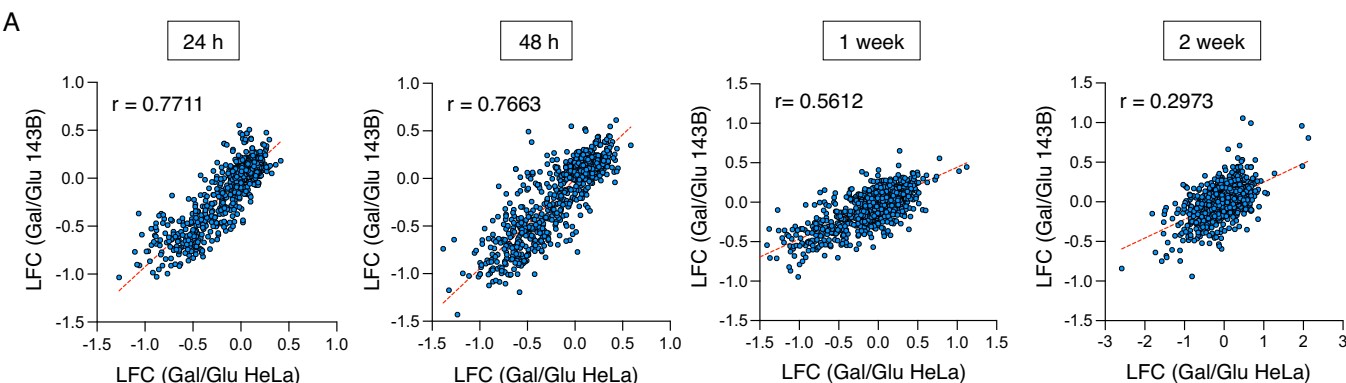

**Figure EV1.  Correlation between HeLa and 143B screening results.**

(A) Correlative analysis of the logarithmic fold change in the galactose/glucose ratio between HeLa and 143B cells at the indicated time points. Source data are available online for this figure.

**A — Dynamics and surveillance**

| Severe Phe. | Dynamics and surveillance |
|---|---|
| 24 hours | ATP5ME CYCS |
| 48 hours | CYCS ATP5ME |
| 1 week | |
| 2 weeks | |

**B — Signaling**

| Severe Phe. | Signaling |
|---|---|
| 24 hours | |
| 48 hours | |
| 1 week | |
| 2 weeks | SLC25A5 |

**D — Transport**

| Severe Phe. | Protein import | Small molecule transport |
|---|---|---|
| 24 hours | UQCRC1 | |
| 48 hours | UQCRC1 | |
| 1 week | UQCRC1 | SLC25A3 SLC25A51 <br> SLC25A19 SLC25A1 |
| 2 weeks | UQCRC1 <br> TOMM22 <br> TIMM21 | SFXN4 SLC25A51 SLC25A1 <br> MPC2 SLC25A28 <br> SLC25A37 SLC25A5 |

**C — MitoCD**

| Severe Phe. | mtDNA maintenance | mtRNA metabolism | mtRNA translation |
|---|---|---|---|
| 24 hours | POLRMT | PRORP POLRMT <br> TRMT10C LRPPRC <br> HSD17B10 | MRPL15 MRPS30 MRPS33 MRPS25 MRPL33 MRPL41 LRPPRC QRSL1 <br> MRPS12 MRPL46 MRPS22 MRPL18 AURKAIP1 MRPL57 NARS2 TUFM <br> YARS2 MRPL4 MRPL23 MARS2 MRPL44 VARS2 MRPS18A |
| 48 hours | TFAM <br> TWNK <br> POLRMT | DDX28 TFAM METTL17 <br> MTPAP LRPPRC <br> FASTKD5 RPUSD4 <br> MRPL12 POLRMT <br> PRORP HSD17B10 | MRPS30 MRPL20 NARS2 MRPL34 MRPL37 MRPL15 QRSL1 TUFM <br> YARS2 MRPL17 MRPL46 MRPL40 MRPS22 LARS2 MRPS18A <br> PTCD3 MRPL12 MRPS33 MRPL4 AARS2 MARS2 VARS2 <br> MRPS6 MRPL23 MRPS23 MRPL57 MRPL38 SARS2 MRPL41 <br> DDX28 MRPL33 PARS2 MRPL44 LRPPRC MRPS12 METTL17 |
| 1 week | | FASTKD5 METTL17 PDE12 <br> GRSF1 MTO1 <br> RMND1 MRM2 | MTRF1 GRSF1 MTIF3 <br> COA3 RMND1 METTL17 <br> GFM2 TACO1 MRM2 |
| 2 weeks | POLDIP2 | PTCD2 MTO1 PDE12 <br> MRM2 GTPBP3 | HEMK1 MRM2 TIMM21 TACO1 |

**E — Metabolism**

| Severe Phe. | Carbohydrate | Lipid | Protein | Nucleotide | Vitamin | Detoxification | Electron carriers | Metal and cofactor |
|---|---|---|---|---|---|---|---|---|
| 24 hours | | HSD17B10 | HSD17B10 | | | | CYC1 <br> CYCS | ISCA1 CYC1 CYCS COX17 COQ4 NDUFS1 <br> NDUFV1 UQCRFS1 NDUFS2 ISCA2 NDUFV2 GLRX5 |
| 48 hours | DLST DLD <br> SDHB OGDH | LIPT2 HSD17B10 <br> LIPT1 | DLST HSD17B10 <br> DLD | | FDX1 | SOD2 | CYCS <br> CYC1 | COQ2 NDUFV1 CYC1 COX17 NDUFS2 ISCA2 GLRX5 <br> CYCS SDHB NFU1 ISCA1 COQ4 NDUFV2 <br> NDUFS8 FDX1 LIAS COX15 UQCRFS1 NDUFS1 |
| 1 week | SDHD OGDH DLD <br> SDHC SDHB PDHB <br> PDHA1 DLAT CS <br> FH DLST SLC25A1 | MECR MCAT SLC25A1 <br> OXSM LIAS LIPT2 <br> ETFB FDX1 <br> GCSH LIPT1 | ETFB <br> GCSH <br> DLST <br> DLD | NME6 | SLC25A19 <br> ETFB <br> FDX1 <br> MTO1 | SOD2 | ETFB <br> CYC1 | SDHD SDHC NDUFS1 NDUFV2 LIAS BOLA3 GLRX5 <br> SLC25A3 HCCS ISCA1 SLC25A51 PDSS2 UQCRFS1 COQ2 <br> COQ5 COX17 SCO1 SDHB CYC1 NFU1 <br> COQ6 NDUFS2 COX15 NUBPL FDX1 COQ7 |
| 2 weeks | SDHB PDHX PDHA1 <br> PDPR DLD PDHB <br> SDHD MDH2 SLC25A1 <br> MPC2 DLAT <br> OGDH CS | LIPT1 ETFA GCSH <br> LIAS MECR ETFB <br> MCAT OXSM LIPT2 <br> ETFDH ECHS1 SLC25A1 <br> HTD2 FDX1 | HIBCH ECHS1 <br> GLUD1 GCSH <br> ETFDH ETFB <br> DLD <br> ETFA | AK3 <br> NME6 <br> SLC25A5 | ETFDH FDX1 <br> RFK GTPBP3 <br> MTO1 ETFB <br> ETFA <br> SLC25A19 | SOD2 | CYC1 <br> ETFDH <br> ETFA <br> ETFB | NUDT19 CYC1 SLC25A51 GLRX5 <br> NDUFS2 COQ2 SLC25A28 <br> SDHB SLC25A37 NFU1 <br> SDHD LIAS FDX1 <br> BOLA3 ETFDH COQ7 |

**F — OXPHOS**

| Severe Phe. | Core subunits | Assembly factors |
|---|---|---|
| 24 hours | COX7C CYC1 NDUFB2 COX5A NDUFC1 UQCRC1 NDUFB4 NDUFA1 <br> NDUFV1 ATP5F1A NDUFB10 CYCS NDUFS2 UQCRB NDUFS1 <br> NDUFB5 NDUFA8 UQCRFS1 NDUFB6 COX4I1 NDUFA2 NDUFS5 <br> NDUFB11 UQCR10 NDUFA6 NDUFB9 NDUFB8 NDUFV2 NDUFA11 | COA6 NDUFAF5 COX16 <br> NDUFAF4 BCS1L COX17 <br> NDUFAF8 COA5 |
| 48 hours | NDUFB7 NDUFV1 NDUFB1 UQCRC1 NDUFB2 NDUFS1 NDUFA1 <br> CYCS NDUFA3 NDUFA6 NDUFA8 UQCRFS1 NDUFC1 NDUFS5 <br> COX5A SDHB UQCRB NDUFB10 COX4I1 NDUFB8 NDUFB4 <br> NDUFS8 NDUFB11 ATP5ME NDUFS2 NDUFV2 NDUFB6 <br> ATP5F1C NDUFA4 CYC1 NDUFA2 NDUFB9 NDUFA11 | NDUFAF3 COA6 COX17 NDUFAF8 COA5 <br> NDUFAF4 DMAC1 BCS1L NDUFAF5 <br> NDUFAF6 FOXRED1 COX15 COX16 |
| 1 week | COX4I1 HCCS UQCR11 NDUFV2 CYC1 NDUFA11 NDUFC1 <br> SDHD UQCRC1 NDUFB10 NDUFA3 NDUFA8 NDUFB2 NDUFA1 <br> SDHC COX6C NDUFS1 SDHB NDUFB8 NDUFA4 <br> NDUFB11 NDUFA10 NDUFB5 NDUFA2 NDUFS5 NDUFB6 <br> NDUFA6 NDUFS2 NDUFB7 NDUFB9 UQCRFS1 NDUFB4 | COA3 COX17 TACO1 NUBPL NDUFAF6 NDUFAF8 <br> PET100 NDUFAF1 DMAC1 COX18 DMAC2 SDHAF1 <br> COX16 ECSIT COA7 LYRM7 NDUFAF4 FOXRED1 <br> COA6 SCO1 COX15 HIGD2A NDUFAF5 <br> TIMMDC1 UQCC2 NDUFAF3 PET117 SURF1 |
| 2 weeks | NDUFB9 SDHD NDUFB4 UQCR11 NDUFB7 NDUFB8 <br> NDUFS2 UQCRC1 NDUFB11 NDUFS5 NDUFA4 <br> NDUFA11 NDUFA3 NDUFA10 NDUFB10 NDUFA1 <br> SDHB CYC1 NDUFB2 NDUFB6 NDUFC1 | PET117 UQCC1 NDUFAF4 FOXRED1 UQCC2 SDHAF1 <br> TIMMDC1 DMAC2 TIMM21 NDUFAF5 TACO1 <br> CEP89 COX16 DMAC1 HIGD2A LYRM7 <br> NDUFAF6 NDUFAF1 COX18 NDUFAF3 SURF1 |

**Figure EV2.** **Genes with positive phenotype in the screening.**

Tables showing the genes that score positive with severe phenotype across different time points for the following functional groups: (**A**) Dynamics and surveillance, (**B**) Signaling, (**C**) Mitochondrial central dogma, (**D**) Transport, (**E**) Metabolism, and (**F**) OXPHOS. Source data are available online for this figure.

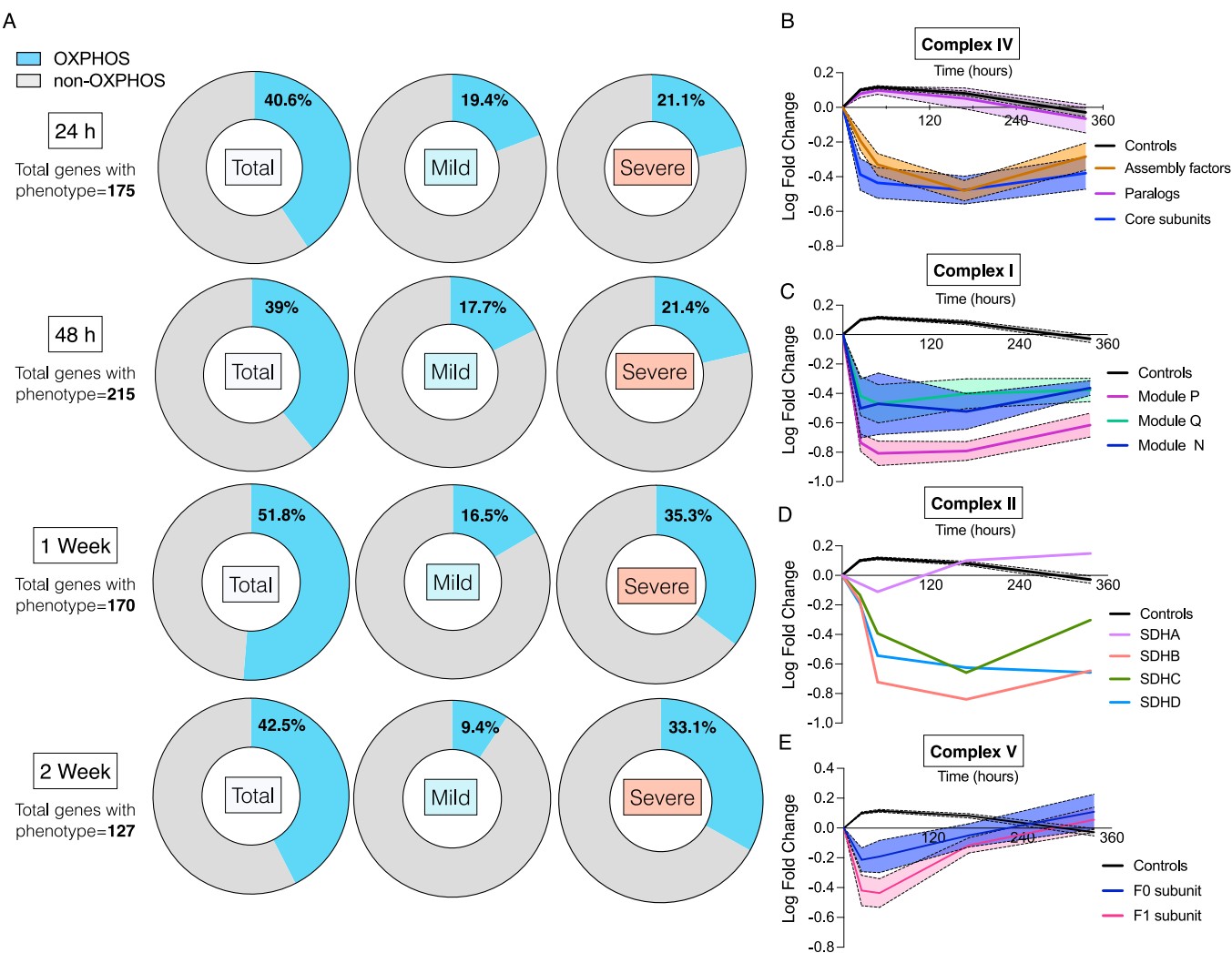

**Figure EV3. Phenotype of OXPHOS genes in the screening.**

(A) Donut charts showing the percentage of OXPHOS versus non-OXPHOS-related genes that scored positive in the screening. The charts illustrate the total number of genes identified, along with those exhibiting both mild and severe phenotypes across various time points. A drop in the fold change in the gal/glu ratio is considered a positive phenotype when it exceeds 20%, with a mild phenotype between 20 and 35%, and a severe phenotype above 35%. (B–E) Temporal dynamic analysis of the logarithmic fold change in the gal/glu ratio of genes involved in the assembly or function of Complexes IV (B), I (C), II (D), and V (E) of the OXPHOS system. Source data are available online for this figure.

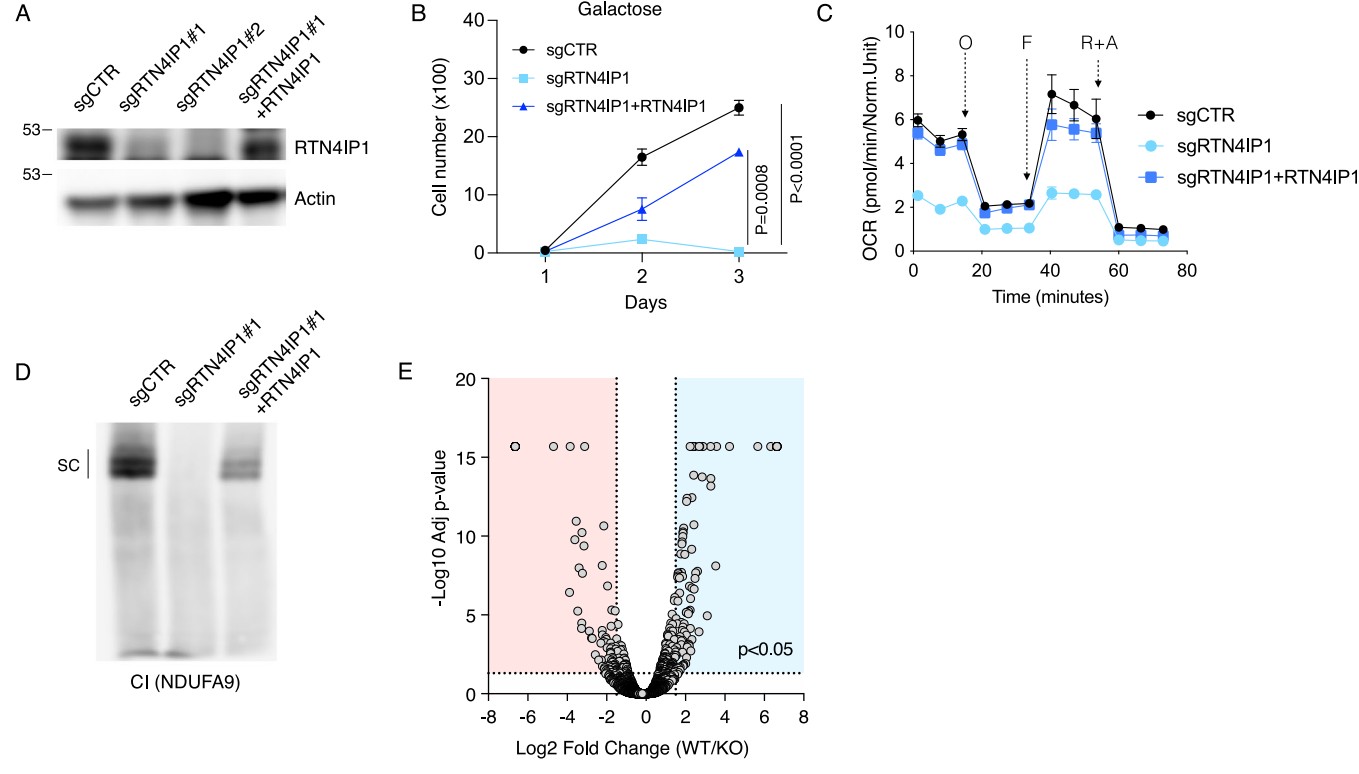

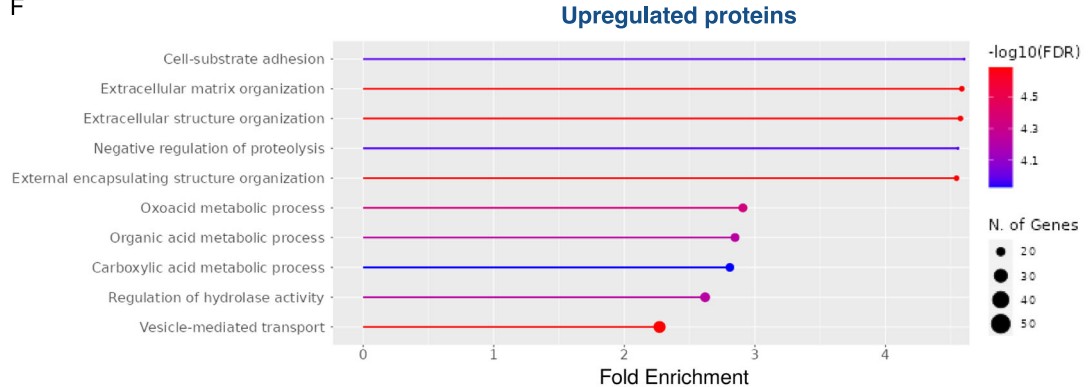

Upregulated proteins

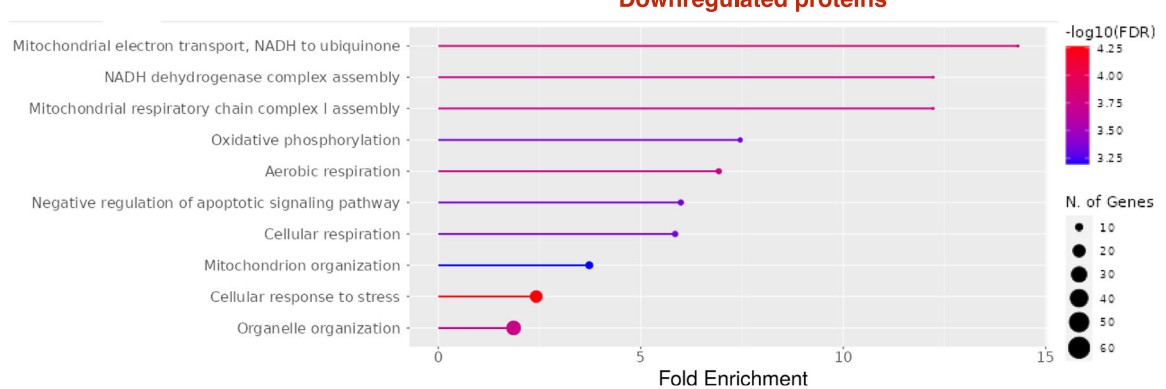

Downregulated proteins

◀ **Figure EV4. Functional rescue and proteomic profiling of RTN4IP1 KO cells.**

(A) Immunodetection of RTN4IP1 in control cells, two stable RTN4IP1 KO clones, and KO cells with ectopic RTN4IP1 expression for rescue. Actin served as a loading control. (B–D) Rescue experiments showing that ectopic RTN4IP1 expression restores galactose-mediated cell survival (B), oxygen consumption (C), and Complex I levels (D) in RTN4IP1 KO cells. (E) Volcano plot illustrating the differential expression of proteins between wild-type (WT) and RTN4IP1 knockout (KO) HeLa cells. (F) Gene Ontology (GO) enrichment analysis showing enriched biological processes, cellular components, and molecular functions from this dataset. Immunoblots shown are representative of >3 independent experiments, and data were presented as means ± SEM. The statistical significance of the differences between groups was determined by a paired two-tailed Student's *t*-test (B, E). To account for multiple comparisons, *p* values were adjusted using the Benjamini–Hochberg false discovery rate (FDR) method. O oligomycin, F FCCP, R + A rotenone plus antimycin a. Source data are available online for this figure.

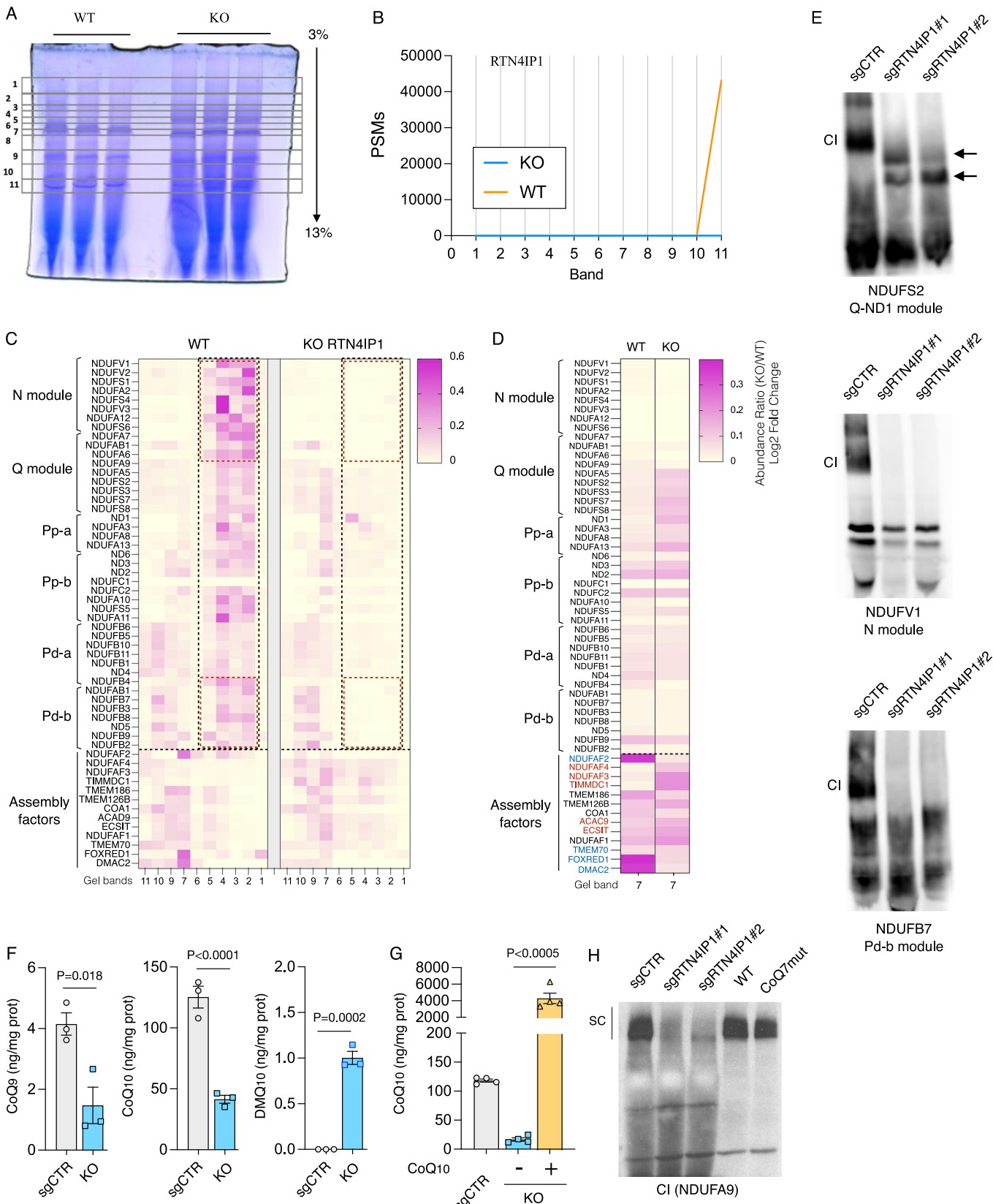

◀ **Figure EV5.  Comprehensive BN-PAGE and proteomic profiling reveal Complex I assembly impairments and CoQ dysregulation in RTN4IP1-deficient cells.**

(A) BN-PAGE of isolated mitochondria from WT RTN4IP1 and KO HeLa cells depicting the eleven bands that were sliced and subjected to proteomic analysis. (B) Peptides corresponding to the RTN4IP1 protein were identified in the previous samples, demonstrating their reduced abundance in the KO samples. (C) Heatmap showing differences in complex I subunits and assembly factors between WT and RTN4IP1 KO HeLa cells. The most notable differences, corresponding to mature complex I and CI-containing supercomplexes, are highlighted with dashed lines. (D) Heatmap focused on band 7, which showed the most striking differences in the accumulation and decrease of several assembly factors. (E) BN-PAGE analysis of DDM-solubilized mitochondria assessing multiple CI modules in WT and RTN4IP1 KO HeLa cells. The Q-ND1 module was identified using NDUFS2, the N-module using NDUFV1, and the Pd-b module using NDUFB7. Arrows indicate accumulated submodules containing NDUFS2 specifically in RTN4IP1 KO cells. (F) Relative level of CoQ9, CoQ10, and DMQ10 in control and RTN4IP1 KO HeLa cells. (G) Restoration of CoQ10 levels in RTN4IP1 KO HeLa cells after supplementing these cells with CoQ10 (20uM) for 1 week. (H) Blue native PAGE (BN-PAGE) showing the levels of CI in control and two stable clones of RTN4IP1 KO HeLa cells, along with healthy human skin fibroblasts (WT) and those carrying a pathogenic variant in the COQ7 gene. Immunoblots shown are representative of >3 independent experiments, and data were presented as means ± SEM. The statistical significance of the differences between groups was determined by a paired two-tailed Student's *t*-test (F) and one-way ANOVA (G). PSMs peptide-spectrum matches. Source data are available online for this figure.

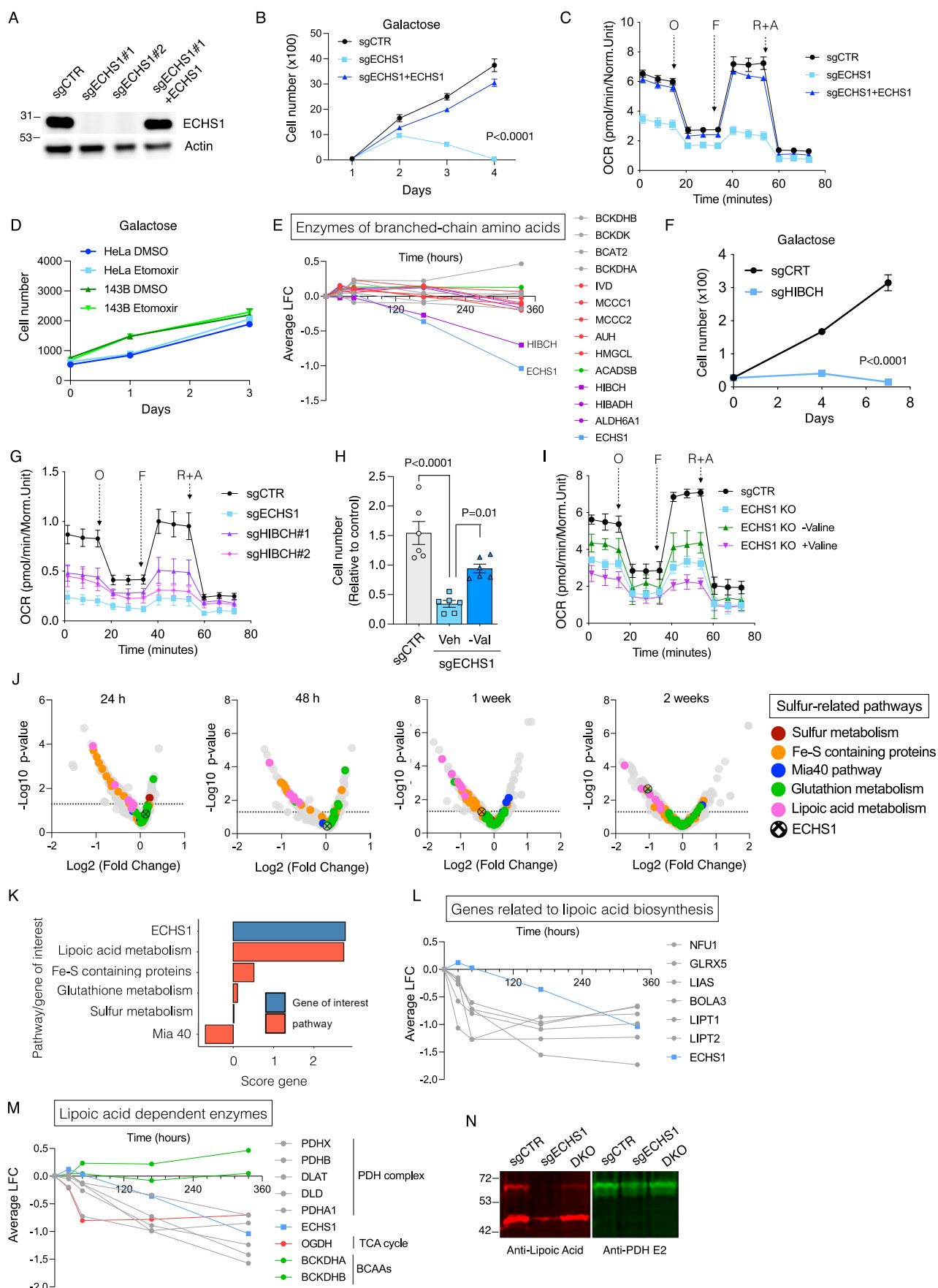

**Figure EV6. ECHS1 deficiency impairs valine catabolism and lipoic acid–dependent enzyme function.**

(A) Immunodetection of ECHS1 in control cells, two stable ECHS1 KO clones, and KO cells with ectopic ECHS1 expression for rescue. Actin served as a loading control. (B, C) Rescue experiments showing that ectopic ECHS1 expression restores galactose-mediated cell survival (B) and oxygen consumption (C) in ECHS1 KO cells. (D) Growth curves depicting the proliferation of HeLa and 143B cells treated with either DMSO or etomoxir. (E) Evolution of the death phenotype (logarithmic fold change in the gal/glu ratio) of all the enzymes involved in the metabolism of branched-chain amino acids in the screening. (F) Cell number under galactose conditions in control and HIBCH KO HeLa cells. (G) Seahorse analysis comparing control, HIBCH KO and ECHS1 KO HeLa cells. (H) Cell number under galactose conditions at 72 h in control HeLa cells and ECHS1 KO HeLa cells in medium with (Veh) and without valine (-Val). (I) Seahorse analysis comparing control and ECHS1 KO cells cultured under conditions without valine and with valine supplementation (4 mM). (J) Volcano plots highlighting the scored values (logarithmic fold change in the gal/glu ratio) of genes involved in sulfur-related pathways and ECHS1 at various time points. (K) Manually curated enrichment analysis of mitochondrial pathways involving thiol groups at two weeks reveals that the ECHS1 score aligns specifically with the lipoic acid pathway. Scores were calculated as logFC * -logPvalue, reflecting the impact of gene knockouts on OXPHOS. The graph depicts the average pathway scores alongside ECHS1. (L, M) Temporal evolution of the death phenotypes (logarithmic fold change in the gal/glut ratio) associated with genes involved in lipoic acid biogenesis (L) and enzymes utilizing lipoic acid as a cofactor (M) and ECHS1. (N) Immunodetection of lipoylated proteins (red color) and the E2 subunit of the PDH complex (green) in mitochondrial extracts of control, ECHS1 KO, and double KO cells for both ECHS1 and ACAD8. Note that the lipoylated signal from PDH E2 is reduced in ECHS1 KO cells. Antibodies were incubated simultaneously on the same membrane and detected with different secondary fluorescent antibodies. Immunoblots shown are representative of >3 independent experiments, and data were presented as means ± SEM. Growth curves are calculated from three replicates and OCR is calculated from five replicates. The statistical significance of the differences between groups was determined by a paired two-tailed Student's $t$-test (B, F) and one-way ANOVA (H). O oligomycin, F FCCP, R + A rotenone plus antimycin a. Source data are available online for this figure.

