## [Peer Review File · EMBO Reports]

Time-resolved mitochondrial screen identifies regulatory components of oxidative metabolism

Eduardo Balsa, Marcos Zamora-Dorta, Sara Laine-Menéndez, David Abia, Pilar González-García, Luis Lopez, Paula Fernández-Montes, Enrique Calvo, Jesús Vázquez, and Jose Antonio Enriquez

Corresponding author(s): Eduardo Balsa (eduardo.balsa@uam.es)

Review Timeline:

Transfer Date:	4th Dec 24
Editorial Decision:	5th Dec 24
Revision Received:	20th Feb 25
Editorial Decision:	21st Mar 25
Revision Received:	26th Mar 25
Accepted:	7th Apr 25

Editor: Deniz Senyilmaz Tiebe

Transaction Report: This manuscript was transferred to EMBO reports following peer review at The EMBO Journal.

Dear Edu,

Thank you for transferring your manuscript to EMBO Reports, which was previously reviewed at another venue.

Having read the manuscript and the referee reports, I would like to invite you to submit a revised manuscript to EMBO Reports as previously communicated. In particular,

- Resolution of complex I assembly assays need to be increased by altering the mitochondrial solubilization as suggested by referee #1.
- Both referees make suggestions regarding data presentation. However, it is not necessary to re-structure/split the manuscript (referee #1).
- Additional rescue experiments and controls are required (referee #2).
- The publications referred by referee #2 need to be cited and discussed in the manuscript, as well as PMID: 35962613, which also shows a role for ECHS1D in OXPHOS complex assembly.
- Conclusions need to be adjusted by taking the concerns of referee #2 regarding cell/tissue specificity of the findings into consideration.

Please address all referee concerns in a complete point-by-point response. Acceptance of the manuscript will depend on a positive outcome of a second round of review. It is EMBO reports policy to allow a single round of major experimental revision only and acceptance or rejection of the manuscript will therefore depend on the completeness of your responses included in the next, final version of the manuscript.

We realize that it is difficult to revise to a specific deadline. In the interest of protecting the conceptual advance provided by the work, we recommend a revision within 3 months. Please discuss the revision progress ahead of this time with me if you require more time to complete the revisions, or if you have questions or comments regarding the revision (also by video chat).

1. A data availability section providing access to data deposited in public databases is missing (where applicable).
2. Your manuscript contains statistics and error bars based on $n=2$. Please use scatter plots in these cases.

You can submit the revision either as a Scientific Report or as a Research Article. For Scientific Reports, the revised manuscript can contain up to 5 main figures and 5 Expanded View figures, and it should not exceed 27000 characters. If the revision leads to a manuscript with more than 5 main figures it will be published as a Research Article. In this case the Results and Discussion section should be separate. If a Scientific Report is submitted, these sections have to be combined. This will help to shorten the manuscript text by eliminating some redundancy that is inevitable when discussing the same experiments twice. In either case, all materials and methods should be included in the main manuscript file.

4) a .docx formatted letter INCLUDING the reviewers' reports and your detailed point-by-point responses to their comments. As part of the EMBO publication's Transparent Editorial Process, EMBO reports publishes online a Review Process File (RPF) to accompany accepted manuscripts. This File will be published in conjunction with your paper and will include the referee reports, your point-by-point response and all pertinent correspondence relating to the manuscript.

<https://www.embopress.org/page/journal/14693178/authorguide#transparentprocess>

5) a complete author checklist, which you can download from our author guidelines

<https://www.embopress.org/page/journal/14693178/authorguide>. Please insert information in the checklist that is also reflected in the manuscript. The completed author checklist will also be part of the RPF.

6) Please note that all corresponding authors are required to supply an ORCID ID for their name upon submission of a revised manuscript (<<https://orcid.org/>>). Please find instructions on how to link your ORCID ID to your account in our manuscript tracking system in our Author guidelines

<<https://www.embopress.org/page/journal/14693178/authorguide#authorshipguidelines>>

7) Before submitting your revision, primary datasets produced in this study need to be deposited in an appropriate public database (see <https://www.embopress.org/page/journal/14693178/authorguide#datadeposition>). Please remember to provide a reviewer password if the datasets are not yet public. The accession numbers and database should be listed in a formal "Data Availability" section placed after Materials & Method (see also

<https://www.embopress.org/page/journal/14693178/authorguide#datadeposition>). Please note that the Data Availability Section is restricted to new primary data that are part of this study. * Note - All links should resolve to a page where the data can be accessed. *

Additional information on source data and instruction on how to label the files are available:

<https://www.embopress.org/page/journal/14693178/authorguide#sourcedata>

9) Our journal encourages inclusion of *data citations in the reference list* to directly cite datasets that were re-used and obtained from public databases. Data citations in the article text are distinct from normal bibliographical citations and should directly link to the database records from which the data can be accessed. In the main text, data citations are formatted as follows: "Data ref: Smith et al, 2001" or "Data ref: NCBI Sequence Read Archive PRJNA342805, 2017". In the Reference list, data citations must be labeled with "[DATASET]". A data reference must provide the database name, accession number/identifiers and a resolvable link to the landing page from which the data can be accessed at the end of the reference. Further instructions are available at <http://www.embopress.org/page/journal/14693178/authorguide#referencesformat>

10) Regarding data quantification (see Figure Legends:

<https://www.embopress.org/page/journal/14693178/authorguide#figureformat>)

12) Please also note our reference format:

13) All Materials and Methods need to be described in the main text using our 'Structured Methods' format, which is required for all research articles. According to this format, the Methods section includes a Reagents and Tools Table (listing key reagents, experimental models, software and relevant equipment and including their sources and relevant identifiers) followed by a Methods and Protocols section describing the methods using a step-by-step protocol format. The aim is to facilitate adoption of the methodologies across labs. More information on how to adhere to this format as well as a downloadable template (.docx) for the Reagents and Tools Table can be found in our author guidelines:

I look forward to seeing a revised version of your manuscript when it is ready. Please let me know if you have questions or comments regarding the revision.

Kind regards,

Deniz

Deniz Senyilmaz Tiebe, PhD
Scientific Editor
EMBO Reports

Point-by-point response to the reviewers' critiques

We would like to thank the editor and the reviewers for the opportunity to revise our manuscript. We appreciate the detailed and constructive feedback provided.

We have carefully considered all the concerns raised by the referees and have undertaken comprehensive revisions to address each of them. We have performed additional experiments requested. Below, we provide a point-by-point response to each of the reviewers' comments, and we have thoroughly revised our manuscript to address all concerns raised.

We believe that these revisions have significantly strengthened our study, and we hope that the reviewers find our responses satisfactory.

REVIEWER COMMENTS

Referee #1:

Major concerns: As much as the presented findings are of great value, my concern is about the manuscript's structure, which possibly hinders the in-depth mechanistic development on each of the two discovered OXPHOS-associated factors. The manuscript contains three parts, with the first devoted to the explanation of the novelty behind the screen's design and the discussion of data. The two subsequent parts explore the role that the discovered factors have in regulating the function of the OXPHOS system. To improve quality and ensure sufficient focus on each studied factor, I would recommend splitting the manuscript into two separate ones to fully expose the two discovered mechanisms. In an optimal scenario, the authors would follow up on each story and resubmit the manuscript focused on the better-characterized aspects. It is still conceivable to consider the revised manuscript with the current structure; however, the following concerns about each part should be addressed.

While we acknowledge the referee's concern regarding the manuscript structure and the suggestion to split it into two separate papers, we believe this is a subjective issue. We consider the current structure appropriate and well-suited for conveying the findings of the manuscript. The design of the screen, followed by the detailed exploration of the two OXPHOS-associated factors, represents a cohesive narrative that aligns with the goals of this work.

When the section focused on RTN4IP1 is concerned, the data point toward its role in the late-stage assembly of CI. The authors observe the accumulation of assembly factors and structural subunits of CI that interact and form subassemblies until the stage when the lack of RTN4IP1 prevents progression to a full complex. Mitochondria solubilized with digitonin are frequently used for complexome profiling. In this specific context, switching to Dodecyl- β -D-maltoside (DDM) detergent for mitochondrial lysis and BN-PAGE analysis would grant better resolution of CI modules (subassemblies) to confirm the accumulation of specific forms (using antibodies against subunits representative of each module). This approach would also allow clear quantification of the degree of loss of each complex, particularly CI, rather than relying on the band that contains CI as part of a supercomplex. For methods allowing the separation of individual complexes, I refer to PMID: 37633273 or PMID: 32242014.

We appreciate the referee's suggestion to use Dodecyl- β -D-maltoside (DDM) for mitochondrial solubilization and BN-PAGE analysis to achieve better resolution of Complex

I modules and to quantify the degree of loss of individual complexes. While we agree that DDM can provide higher resolution in separating CI modules, we would like to highlight some concerns and limitations associated with the use of such a strong detergent. It is well-documented, including in our own work (PMID: 22902835), that DDM is an artifact-prone detergent that can disrupt protein-protein interactions and remove loosely bound proteins. Consequently, results obtained using DDM should be interpreted cautiously, as they might not accurately reflect the native state of protein complexes in mitochondria.

Taking all this into consideration and in response to the referee's suggestion, we performed experiments using DDM and specific antibodies against various CI modules. These analyses, now shown in EV5E, further confirm the accumulation of Q-ND1 module subassemblies, as predicted by the referee. However, no accumulation was observed for the N-module or Pd-b module. Overall, these findings are consistent with our complexome profiling data.

Figure 5 contains meaningful data; however, their essence is compacted into a figure (A) that is not easy to read (compacted overlapping curves). Complexome profiling data would benefit from being displayed as a heatmap, with each detected subunit shown separately (as in supplementary figure). Trends observed for modules could then be juxtaposed. If the manuscript focused solely on the screen and RTN4IP1, a separate figure could then be devoted to investigating its role in CoQ synthesis. Data from supplementary figure 5 could also be highlighted within the main figures.

We appreciate the detailed examination of Figure 5C and its presentation, as well as the insights into optimizing the visualization of complexome profiling data. This figure was carefully designed with the aim of providing a comprehensive and concise summary of the complexome profiling data in a way that conveys the essential findings at a single glance. We believe that the overlapping curves, while compact, are integral to presenting the interconnected nature of the dataset and allow the broader trends to emerge clearly within the context of the figure.

To complement this integrative visualization, the supplementary figures provide a more detailed breakdown of the data, including individual subunit trends and finer aspects of the dataset.

The finding of ECHS1 and its role in valine metabolism regulating OXPHOS is of high importance and interest. Yet, this line of investigation has not been sufficiently developed. The obtained findings, instead of being fully explored, lose potential impact when contained in one short figure. BNPAGE analysis using digitonin does indicate a decrease in supercomplex levels. Using DDM-solubilized mitochondria instead could help to determine the effect of ECHS1 loss on each complex level. Considering that in fact the loss of supercomplex upon ECHS1 knockout is significant, it might also be worthwhile to check the migration of ECHS1 on BNPAGE (digitonin solubilized mitochondria), given that FAO enzymes have previously been shown to associate with supercomplexes. This would help determine whether the loss of ECHS1 has any structural effect on supercomplex abundance and thus, on respiration.

We thank the referee for their insightful comments regarding the role of ECHS1 in regulating OXPHOS and its potential association with supercomplexes. While we acknowledge the suggestion to use DDM-solubilized mitochondria for further analysis, the levels of supercomplexes in our study show only minimal changes upon ECHS1 loss. We do not

consider this minor phenotype to warrant further exploration in the context of this manuscript.

Additionally, while the proposed BNPAGE analysis to examine ECHS1 migration and structural effects is interesting, it falls outside the current focus of our study, which prioritizes the metabolic and functional consequences of ECHS1 loss.

The ECHS1 sub-story should, however, be mostly extended by additional approaches assessing the consequences of disrupting valine metabolism. This part would greatly benefit from devising experiments to demonstrate increased levels of methacrylyl-CoA and 3-hydroxyisobutyryl-CoA. The link between these metabolites and the lipoic acid biosynthesis pathway is interesting but not sufficiently documented. The lipoylation defect is shown, as well as the reduced PDH activity. However, upstream of these effects, evidence on adducts within particular enzymes of the lipoic acid synthesis pathway is lacking. Overall, the study lacks direct evidence that the aforementioned metabolites indeed modify targets in the proposed manner. Efforts to strengthen conclusions on valine metabolites modifying lipoic acid biosynthesis factors would constitute a major point to be addressed by the authors as part of the revision.

We thank the referee for their thoughtful suggestions regarding the ECHS1-related findings and the proposed additional experiments to further explore the role of valine metabolism and its link to lipoic acid biosynthesis. While we acknowledge the value of these approaches, they fall outside the current scope of our study.

Minor concerns: Regarding the conclusions from the part devoted to the CRISPR/Cas9 screen, I would draw attention to the sentence explaining why a "more aggressive" phenotype is observed for genes encoding P-module subunits. I would refrain from comparing the importance of each CI module's function but would possibly direct attention toward the P-module's role in the biogenesis of other complexes. Its lack might have a more severe phenotype due to disturbed assembly of CIII and CIV (exemplary reference: PMID: PMC3318979).

Thank you for the suggestion. We have incorporated this point into the revised manuscript. Specifically, we now discuss the potential role of the P module in the biogenesis and assembly of other respiratory chain complexes, such as CIII and CIV, to explain the more severe phenotype observed upon its disruption. This addition is supported by the reference provided (PMCID: PMC3318979) and has been included to strengthen the interpretation of our findings.

For figures 4 and 5 containing BNPAGE images, images obtained with shorter exposure times should be added, particularly for CIV and CII. For Figure 5E, quantification of the CoQ effect should be included to substantiate the conclusions. Figure 5D contains a typo: "ration" instead of "ratio."

Thank you for pointing out the typo in Figure 5D; "ration" has been corrected to "ratio" in the revised version of the manuscript. Regarding the suggestion to include quantification for the CoQ effect in Figure 5E, we appreciate the feedback but believe that quantification is unnecessary, as it is visually evident that the levels of supercomplexes are not restored.

A more thorough labeling of figure elements for improved readability should be included.

Figure 5C should include short labels explaining what the displayed values mean (for example, the fate of CI AF upon KO, without needing to check the figure description). Similarly, figure 4D should be more self-explanatory and indicate the source of the analyzed data. The heatmap in figure 4G should contain information on how abundance is defined (raw values, fold change, or log₂ fold change?). Other figures should be revised in a similar fashion, improving labels and readability.

Thank you for your suggestions regarding figure labeling and readability. We acknowledge that these are subjective matters, and some revisions have been made to improve clarity in the updated manuscript. However, we believe that the current presentation adequately conveys the necessary information while maintaining a balance between detail and readability. Information in the heatmap figures now included log₂ fold change.

Referee #2:

Zamora-Dorta et al employed a time-course CRISPR/Cas9 screen using guides for known nuclear-encoded mitochondrial genes where the enrichment (or depletion) of genes in galactose vs glucose was used to identify novel regulators of mitochondrial OXPHOS. In doing so, they identified RTN4IP1 and ECHS1 as genes impacting OXPHOS function. They performed followup studies on each of these, including generation of KOs and various functional studies. The authors concluded that RTN4IP1 is involved in both CI assembly and CoQ biosynthesis, thus likely representing identification of a novel CI assembly factor. ECHS1 has an established role in valine and FAO metabolism and the authors find that they can rescue ECHS1 deficiency by valine restriction or knockout of an upstream enzyme in BCAA metabolism, concluding that the major defect in cells comes from impacted BCAA rather than FAO metabolism. Similar conclusions have been made in clinical studies involving valine restriction in humans. (eg. PMID 36064416, 33139125). This lessens the novelty, though I note that the authors provide more in depth molecular detail than the clinical studies. A recent study also presented a ECHS1/ACAD8 double knockout (PMID: 37309295) but in their case this did not rescue lipoylation of PDH complex components on its own, and the authors required additional use of small molecule inhibitors. Together these results suggest to me that there is a lot of cell line/tissue specificity to these processes, which is noted by the authors in the discussion, but in my view this means the conclusion should be more cautiously presented.

We thank the reviewer for bringing this recent study to our attention. We have included the reference (PMID: 37309295) in the revised manuscript and have updated the discussion accordingly. We agree with the reviewer that these variations highlight the complexity of the processes involved, and we have emphasized this point in the new discussion section.

Overall this is a well designed and technically sound study. Aside from my comments on novelty and strength of conclusions, I have the following major comments/concerns:

- The data showing the reduction of CI species via BN-PAGE in the RTN4IP1 KO supports this protein being critical to CI assembly, however, the authors do not include a rescue/complementation of the RTN4IP1 KO line to confirm the restoration of CI and OXPHOS function via BN-PAGE and Seahorse analysis. This is crucial for establishing that RTN4IP1 is a bona fide CI assembly factor. The conclusion around the specific role of this protein in assembly of the Q-ND1 module is less sound and the authors should do an immunoprecipitation to demonstrate a direct interaction with the relevant subunits. Both

issues could be resolved in a single set of experiments where eg a FLAG or HA tagged RTN4IP1 is used both for complementation and immunoprecipitation.

In response to the concern regarding the complementation of the RTN4IP1 knockout (KO) line, we have performed rescue experiments. These experiments confirm the critical role of RTN4IP1 in the specific assembly of Complex I, as shown by the restoration of CI species in BN-PAGE analysis. We believe this data provides strong support for the role of RTN4IP1 in CI assembly and mitochondrial function. Now included in Figures EV4A-D.

Regarding the suggestion to investigate the direct interaction between RTN4IP1 and relevant CI subunits through immunoprecipitation (IP), we appreciate the reviewer's insightful recommendation. While we agree that such experiments could provide valuable information, we believe that exploring this mechanism in detail would introduce additional complexity and is somewhat beyond the scope of our current investigation. Our current focus is on the broader functional role of RTN4IP1 in CI assembly, which we believe is well supported by the data presented.

- The authors do not explain how/if they validated the KO cell lines as there is no SDS-PAGE/WB analysis or sequencing results that show specific loss of the protein or genomic edits. One of these should be shown for each KO.

We have now included validation of the knockout (KO) cell lines in the revised manuscript. Western blot analyses that confirm the specific loss of RTN4IP1 and ECHS1 in the KO lines (and rescue experiments) are now presented in the revised version of the manuscript to ensure clarity and rigor in the validation of the KO lines. Now included in Figures EV4A and EV6A-C.

Minor concerns that should be addressed if possible

- I found the way the screen results were presented a little confusing, especially as the authors switch from Log FC to a linear ratio part-way through the manuscript. For simplicity I suggest to show all plots as Log FC instead of ratio. Also, why does NDUFB5 in Fig 3 have an abundance ratio >1 and in Fig 4D it is reduced, ~0.5?

Thank you for your observation regarding the presentation of the screen results. While we appreciate your feedback, we believe the data is presented in a clear and accurate manner. There seems to be a discrepancy because Figure 3 does not show the presence of NDUFB5.

- The matrix containing the proteomics and complexome data should be made available as SI
We have now included the matrix containing the proteomics of RTN4IP1 WT/Ko cells and complexome data as a supplementary file (SI) for transparency and ease of access.

- Labelling of Supplementary figures should be consistent and as Line 126, Figure S1A or Supplementary Figure 1A, subsequent labels are as Figure 2SC (example in line 168 and throughout the manuscript).

It has been corrected

- Line 179, in the exception list of genes, does early/late detection of assembly factors correlate with disease severity or prevalence?

No

- Line 189, The authors mention that they investigate the temporal fitness of each OXPHOS complex but it does not comment on complex III, why?

However, we did not specifically address complex III in the temporal fitness analysis, as it is more challenging to partition complex III into distinct functional modules compared to other OXPHOS complexes.

- Line 238 needs reference.

It has now been included

- Fig S5.C, why is column 8 missing? and label the gel markers (kDa) on all complexome figures.

Unfortunately, this band was lost during sample processing.

- Line 868, 'al' should be all?

This typo has been addressed

Dear Edu,

Thank you for submitting your revised manuscript. It has now been seen by both of the original referees.

As you can see, referees find that the study is significantly improved during revision and recommend publication. However, I need you to address the points below before I can accept the manuscript.

- Please address the remaining minor concerns of referees #1 and #2 by textual changes.
- As per our guidelines, please add a 'Data Availability Section', where you state that no data were deposited in a public database.
- Please rename the 'Declaration of Interests' section as 'Disclosure And Competing Interests Statement'.
- We note a name discrepancy - Eduardo Balsa in the manuscript vs. Eduardo Balsa Martinez in the manuscript tracking system.
- Please remove the 'Author Contributions' section from the manuscript text.
- As per our format requirements, in the reference list, citations should be listed in alphabetical order and then chronologically, with the authors' surnames and initials inverted; where there are more than 10 authors on a paper, 10 will be listed, followed by 'et al.'. Please see <https://www.embopress.org/page/journal/14693178/authorguide#referencesformat>
- We note that the Author Checklist is missing responses in the pull down menu (column D).
- We note the following regarding the funding information: missing in the manuscript tracking system - Consejo Superior de Investigaciones Científicas and Universidad Autónoma de Madrid, and is a Severo Ochoa Center of Excellence (grant CEX2021-001154-S) funded by MICIN/AEI/10.13039/501100011033; FPI-UAM PhD fellowship; Margarita Salas-UAM postdoctoral fellowship; European Union "NextGenerationEU"/Plan de Recuperación Transformación y Resiliencia -PRTR; CIBERFES (CB16/10/00282) funded by ISCIII; FPU21/06416; Instituto de Salud Carlos III (ISCIII); the Ministerio de Ciencia e Innovación (MICINN); Pro CNIC Foundation, and is a Severo Ochoa Center of Excellence (grant CEX2020-001041-S funded by MICIN/AEI/10.13039/501100011033); all these need to be entered via More Funders option and the Comments box should not be used.
- We note that there are two Excel files uploaded as Table 1 and Table 2; these are datasets and need to be updated to Dataset EV1 and Dataset EV2 in all places (source file names, titles in the manuscript tracking system, manuscript callouts). Similarly, Data set proteomics Complexome and Data Set whole cell Proteomics should be renamed as Dataset EV3 and Dataset EV4, respectively.
- Please remove the Reagents and Tools table from the manuscript text and submit it as a separate word file.
- Please provide source data as requested by our source data coordinator Dr. Hannah Sonntag in her email dated 13.12.2024. Also, please fill in the source data checklist attached to the same email and submit it along with the source data.
- Summary should be renamed as Abstract.
- Inclusion And Diversity statement/section needs to be removed from the manuscript.
- The manuscript sections should be in the following order: Title page - Abstract & Keywords - Introduction - Results - Discussion - Methods - Data Availability - Acknowledgments - Disclosure Statement & Competing Interests - References - Figure Legends - (Main Tables with legends if applicable) - Expanded View Figure Legends.
- Our production/data editors have asked you to clarify several points in the figure legends - Figure Legends (main + EV):
 - o Please note that the figure EV6 is mislabeled as figure EV5 in the legends of the manuscript. This needs to be rectified.
 - o Please note that the exact p values are not provided in the legends of figures 1C, 4A, B; 6F, H, I; EV4 B; EV5 F, G; EV6 B, F, H.
 - o Please indicate the statistical test used for data analysis in the legends of figures 1C, 4B, 6D, EV4 E.
- Papers published in EMBO Reports include a 'synopsis' and 'bullet points' to further enhance discoverability. Both are displayed on the html version of the paper and are freely accessible to all readers. The synopsis includes a short standfirst summarizing the study in 1 or 2 sentences (max 35 words) that summarize the paper and are provided by the authors and streamlined by the handling editor. I would therefore ask you to include your synopsis blurb and 3-5 bullet points listing the key experimental findings.
- In addition, please provide an image for the synopsis. This image should provide a rapid overview of the question addressed in the study but still needs to be kept fairly modest since the image size cannot exceed 550 (width) x 300-600 (height) pixels.

Thank you again for giving us to consider your manuscript for EMBO Reports, I look forward to your minor revision.

Kind regards,

Deniz

--

Deniz Senyilmaz Tiebe, PhD
Senior Scientific Editor
EMBO Reports

Referee #1:

Thank you for addressing our major concerns regarding validation of the knockout cell lines and lack of complementation experiments. It was great to see multiple assays being used. In saying this I note that the details of these experiments are not included in the methods, which should be corrected to the satisfaction of the editor.

While I'm willing to accept their argument that it is out of scope, I still think it was a missed opportunity to not do RTN4IP1 immunoprecipitation, especially given they included DLAT IP data in the ECHS1 part of the story.

My minor comments were suitably addressed. Finally, I agree with the authors argument against Reviewer 1 concerns on manuscript structure and the potential for DDM to introduce artifacts on BN-PAGE analysis, given the reality that these complexes likely interact with each other during assembly.

Referee #2:

In the study entitled Time-resolved mitochondrial-focused screening identifies regulatory components of oxidative metabolism by Zamora-Dorta et al., authors performed a time-resolved genetic screen with the use of a CRISPR/Cas9 loss-of-function library targeting nuclear-encoded mitochondrial genes and galactose-containing media that selects against OXPHOS-deficient cells. Through this, they generated a catalogue of genes essential for OXPHOS function, with particular focus on novel players whose loss-of-function effects reveal itself in later time-points (1-2 weeks) than usually utilised in similar studies. Authors then focused on functional characterisation of two factors - RTN4IP1 and ECHS1 - associated with OXPHOS. Their mutations cause mitochondrial diseases, yet encoded proteins themselves are not known OXPHOS structural or assembly factors. The findings are significant and bring valuable insights into the mechanism of CI assembly and post-translation modification of metabolic machinery dependent on BCAA catabolism, respectively. The value of this study highlights the fact that a still significant proportion of mitochondrial proteome misses exact functional annotation.

Addressing the reviewer's questions received during the review process for EMBO J notably improved the manuscript. Considering the format preferences of EMBO Reports, I find the manuscript by Zamora-Dorta et al. appropriate for publication. The functional insights that emerged from the presented screen of factors regulating oxidative metabolism are suitable and sufficient for the journal.

Minor comments are as below:

Lines 102-103: In the summary of the findings presented in the introduction, I suggest to specify the role of ECHS1 in regulating oxidative metabolism. Currently, it is only highlighted that it is independent of the canonical function.

Lines 392-395: It would be recommended that authors provide a more thorough explanation of how they approached the task of identifying pathways that rely on thiol groups for their function. The reference/source of this annotation is missing.

All editorial and formatting issues were resolved by the authors.

Prof. Eduardo Balsa
Universidad Autonoma de Madrid
Nicolar Cabrera 1
Madrid, Madrid 28049
Spain

Dear Edu,

Thank you for submitting your revised manuscript. I have now looked at everything and all is fine. Therefore, I am very pleased to accept your manuscript for publication in EMBO Reports.

Congratulations on a nice work!

Before we can export your manuscript to our production team, I need your input on two minor points:

1. I propose a minor change in the title (as below). Please take a look and confirm, or feel free to propose further changes:

Time-resolved mitochondrial screen identifies regulatory components of oxidative metabolism

2. · Design-wise, the synopsis image looks very nice. However, when it is resized according to the format requirements, the labels become too small to read (please see attached). Please provide a synopsis image with larger labels.

Thank you.

Kind regards,

Deniz

--

Deniz Senyilmaz Tiebe, PhD
Senior Scientific Editor
EMBO Reports

--
